# MicroRNA and Metabolic Profiling of a Primary Ovarian Neuroendocrine Carcinoma Pulmonary-Type Reveals a High Degree of Similarity with Small Cell Lung Cancer

**DOI:** 10.3390/ncrna8050064

**Published:** 2022-09-25

**Authors:** Stefano Miglietta, Giulia Girolimetti, Lorena Marchio, Manuela Sollazzo, Noemi Laprovitera, Sara Coluccelli, Dario De Biase, Antonio De Leo, Donatella Santini, Ivana Kurelac, Luisa Iommarini, Anna Ghelli, Davide Campana, Manuela Ferracin, Anna Myriam Perrone, Giuseppe Gasparre, Anna Maria Porcelli

**Affiliations:** 1Department of Pharmacy and Biotechnology (FABIT), University of Bologna, 40126 Bologna, Italy; 2Center for Applied Biomedical Research (CRBA), University of Bologna, 40138 Bologna, Italy; 3Department of Medical and Surgical Sciences (DIMEC), University of Bologna, 40138 Bologna, Italy; 4Unit of Transplant immunobiology and Advanced Cell Therapy, IRCCS Azienda Ospedaliero-Universitaria di Bologna, 40138 Bologna, Italy; 5Department of Experimental Diagnostic and Specialized Medicine (DIMES), University of Bologna, 40138 Bologna, Italy; 6Solid Tumor Molecular Pathology Laboratory, IRCCS Azienda Ospedaliero-Universitaria di Bologna, 40138 Bologna, Italy; 7Pathology Unit, IRCCS Azienda Ospedaliero-Universitaria di Bologna, 40138 Bologna, Italy; 8Centro Studi E Ricerca Sulle Neoplasie Ginecologiche (CSR), University of Bologna, 40138 Bologna, Italy; 9Division of Medical Oncology, IRCCS Azienda Ospedaliero-Universitaria di Bologna, 40138 Bologna, Italy; 10Division of Oncologic Gynecology, IRCCS Azienda Ospedaliero-Universitaria di Bologna, 40138 Bologna, Italy; 11Interdepartmental Center of Industrial Research (CIRI) Life Science and Health Technologies, University of Bologna, 40064 Ozzano dell’Emilia, Italy

**Keywords:** microRNA, small cell neuroendocrine carcinoma, ovarian carcinoma, gynecological cancers, mTOR, cancer metabolism

## Abstract

Small cell neuroendocrine carcinoma is most frequently found in the lung (SCLC), but it has been also reported, albeit with a very low incidence, in the ovary. Here, we analyze a case of primary small cell carcinoma of the ovary of pulmonary type (SCCOPT), a rare and aggressive tumor with poor prognosis, whose biology and molecular features have not yet been thoroughly investigated. The patient affected by SCCOPT had a residual tumor following chemotherapy which displayed pronounced similarity with neuroendocrine tumors and lung cancer in terms of its microRNA expression profile and mTOR-downstream activation. By analyzing the metabolic markers of the neoplastic lesion, we established a likely glycolytic signature. In conclusion, this in-depth characterization of SCCOPT could be useful for future diagnoses, possibly aided by microRNA profiling, allowing clinicians to adopt the most appropriate therapeutic strategy.

## 1. Introduction

Neuroendocrine tumors of the female genital tract are rare neoplasms which represent less than 2% of all gynecological cancers [1]. The 2020 World Health Organization (WHO) classification revolutionized the terminology of gynecological neuroendocrine tumors, dividing them into two main groups: well-differentiated neuroendocrine tumors (NETs)—including designated carcinoid tumors in the ovary—and poorly differentiated neuroendocrine carcinomas (NECs). This dichotomous pathological categorization is supported by molecular evidence at specific anatomical sites, as well as by clinical, epidemiological, histological, and prognostic differences [2,3]. In the ovary, low-grade neoplasms predominate and represent the most common primary neuroendocrine neoplasm in the female genital tract, arising mainly within teratomas, especially dermoid cysts (mature cystic teratomas) and displaying benign behavior. In contrast, primary ovarian NECs are exceptional entities. Ovarian neuroendocrine carcinomas or small cell carcinoma of the ovary are categorized into two types in the literature: hypercalcemic and pulmonary [4]. Small-cell carcinoma of the ovary pulmonary type (SCCOPT) is a very rare form of neuroendocrine carcinoma, which is morphologically identical to small-cell lung cancer (SCLC), and is therefore characterized by highly aggressive behavior and poor outcome [5]. Treatment of SCCOPT is not standardized, so each case should be approached in a multidisciplinary manner in tertiary centers. Therapy may include cytoreductive surgery, radiotherapy and chemotherapy with regimens commonly used in SCLC, e.g., Cisplatin and Etoposide [6,7]. Due to its rarity, investigations of the biological and molecular features of SCCOPT are lacking. Genetic alterations in *TP53* and *BRCA2* have been shown to have few functional implications for therapeutic choices [8]. Here, in seeking to identify indications that are suggestive of common characteristics of SCLC which may orient and justify therapeutic strategies, we perform microRNA (miRNA) profiling and apply molecular biology/genetics techniques to determine the metabolic features of a case of SCCOPT.

## 2. Results

### 2.1. Imaging, Morphological and Immunohistochemical Analyses Depict a Rare Ovarian Neuroendocrine Carcinoma Pulmonary-Type

A 78-year-old woman was admitted to the gynecologic oncology outpatient clinic in January 2017 for asthenia, nausea, weight loss, and pelvic swelling. Trans-vaginal ultrasonography revealed a left adnexal solid mass of 7 cm, with irregular margins and strong blood flow (CS 3), connected to the pelvic floor and the ileum (Figure 1A,B). The right ovary and perimetrium presented hypoechoic nodules. Pelvic free fluid was present, and the Douglas peritoneum was thickened. Trans-abdominal ultrasonography evidenced a 4 cm para-aortic metastatic lymph node. A computed tomography (CT) scan confirmed the sonographic findings. The patient, due to her poor general condition, was unfit for surgery. Therefore, a transvaginal ultrasound-guided core needle biopsy of the left adnexal mass was performed. The pathological examination showed a high-grade carcinoma which was compatible with a gynecological origin.

The patient received six cycles of neoadjuvant chemotherapy with Carboplatin AUC 5 and Paclitaxel 175 mg/m(2), ending in July 2017. A CT scan showed a reduction of the left ovarian mass (29 × 22 mm), no signs of peritoneal carcinomatosis after neoadjuvant chemotherapy and improved general conditions (Figure 1C). The patient then became eligible for surgery, and in September 2017 laparotomic hysterectomy, bilateral salpingo-oophorectomy, peritoneal washing, omentectomy, resection of peritoneal pararectal nodules and peritoneal biopsies were performed. The left ovary had transformed into an irregular neoplastic mass 6 cm in width, attached to the rectum. Additionally, three peritoneal pararectal nodules of 15–60 mm in size were detected. Following surgery, there was no macroscopic residual disease (R0). There were no post-operative complications, and the patient was discharged from the Gynecologic Oncology clinic after five days. The patient died from disease progression in January 2018. An examination of the left ovarian mass revealed a 6 cm lesion with yellowish cut surface, solid and partially cystic components and areas of necrosis and hemorrhage. Microscopically, the tumor was composed of small cells arranged in sheets and closely packed nests. The neoplastic cells showed hyperchromatic nuclei with inconspicuous nucleoli and scant cytoplasm (Figure 1D). Mitotic figures were numerous, some with atypical features (Figure 1D). Endometrioid, serous, mucinous and teratomatous elements were absent. Lymphovascular space invasion was diffuse. The tumor involved both ovaries and the left fallopian tube, with some neoplastic peritoneal nodules. A routine diagnostic examination ruled out a tubo-ovarian high-grade serous carcinoma (HGSC) due to immunohistochemistry staining showing negative for WT-1, PAX-8 and estrogen receptor/progesterone receptor [3,9] (not shown). On the other hand, chromogranin diffuse expression (Figure 1E) suggested neuroendocrine differentiation [10]. Tumor cells were also positive for TTF-1 and p53 (Figure 1F,G). These morphological and immunohistochemical findings were similar to those of SCLC, but a metastatic origin was excluded due to the lack of lung lesions in the CT scan, pointing to a diagnosis of SCCOPT. To characterize the tumor mutation profile, primary ovarian cancer, peritoneal metastasis and non-tumor tissue were analyzed by next-generation sequencing (NGS). The analysis allowed to detect the *TP53* p.Y163C (c.488A > G, exon 5) missense mutation in all cancer specimens, correlating to a strong and diffuse p53 overexpression (Figure 1G) and confirming the already reported frequent alteration of *TP53* in SCCOPTs. The coverage (total reads) for each analyzed specimen and the VAF (Variant Allele Frequency) were as follows: primary tumor 1726× (*TP53* c.488A > G VAF: 92%); peritoneal metastasis 1141× (*TP53* c.488A > G VAF: 82.7%); non-neoplastic tissue 4774× (*TP53* c.488A > G variant not detected). Such *TP53* mutation is classified as “pathogenic” according to American College of Medical Genetics (ACMG) classification and the ClinVar archive (https://varsome.com, accessed on 27 July 2022). No alterations were detected in the other analyzed genetic regions.

### 2.2. MicroRNA Profiling of SCCOPT Reveals High Similarity with Neuroendocrine and Lung Cancers

To understand whether the molecular profile of SCCOPT was similar to the most common small cell neuroendocrine carcinomas that arise in the lung, we applied a molecular test based on 81 miRNA profiling that was developed to identify the most probable primary tumor among 17 primary tumor classes [11]. Indeed, some miRNAs display a tumor-specific expression profile, which could be used to infer the origin or similarities among neoplasms. We analyzed the miRNA expression profile of our SCCOPT case and compared it with the 17 tumor types (see Appendix A). We performed a clustering analysis, using Pearson correlation as a similarity measure, revealing that “neuroendocrine” tumors had the most similar expression profile (Figure 2). Furthermore, we applied a recently developed predictive algorithm to identify the most probable tumor type for unknown samples [11]. The miRNA-based analysis for this patient pointed to “neuroendocrine” and “lung cancer” as the most probable tumor types, with probabilities of 40% and 20%, respectively. As an additional observation, we noticed that miRNA-34b-3p and miR-485-5p were expressed at very low levels in the SCCOPT sample, as also reported in SCLC [12]. One possible explanation is that the miR-34 family is transcribed by p53, and its downregulation is associated with *TP53* mutations [13]. A list of the most expressed miRNAs in the SCOOPT panel is reported in Table 1. Among these miRNAs, we detected miR-375, miR-141-3p, miR-200c-3p, miR-16-5p and miR-103a-3p. MiR-375 is one of the main regulators of endodermal differentiation and was first discovered as the main regulator of insulin secretion in pancreatic cells [14]. It is widely expressed in endocrine and neuroendocrine cells (see also Appendix A) an evidence that further links our SCCOPT case to neuroendocrine carcinomas. MiR-141-3p and miR-200 families are upregulated in prostate cancers and regulate the metastatic process in many tumor types. In ovarian cancer, the overexpression of the miR-200 family has been associated with a mesothelial-to-epithelial transition and a more aggressive phenotype [15].

### 2.3. SCCOPT Is Characterized by mTOR Downstream Activation and Glycolytic Profile

SCLC is characterized by aggressive growth and poor prognosis, and no molecular targeted drugs have shown clinical efficacy [16]. In these cancers, the PI3K/Akt/mTOR pathway plays a central role in survival, proliferation, migration and metabolic rewiring [17], wherein it was reported to be hyperactivated [18]. In this frame, mTORC1-downstream p70 ribosomal S6 kinase (S6K) has been shown to be upregulated in lung cancer and is thus considered a prognostic marker [19].

To identify molecular and metabolic similarities between our case and SCLC, we investigated the levels and the activation of multiple players in SCCOPT and peritoneal metastasis (PM) using non-cancer ovarian tissue (NC-OV) and a case of high-grade serous carcinoma (HGSC) as controls. Indeed, an increased level of phosphorylated S6K (P-S6K) was observed in SCCOPT and in PM compared with the controls (Figure 3A), indicating the activation of such pathway and highlighting the similarity of this case with SCLCs.

Additionally, we analyzed pyruvate kinase M (PKM), a glycolytic enzyme expressed as splice variants encoding the PKM1 and PKM2 isoforms that convert phosphoenolpyruvate (PEP) to pyruvate during glycolysis. Unlike constitutively active PKM1, PKM2 is considered a hallmark of cancer [20], since most cancer cells predominantly express this isoform, whose activity supports the Warburg effect and metabolic rewiring [21]. Indeed, PKM2 activity maintains the glycolytic flux at a lower rate and limits glucose oxidation, favoring the generation of glycolytic metabolic intermediates with subsequent activation of the pentose phosphate pathway [22]. Interestingly, SCLCs preferentially express the PKM1 isoform, which is required for cancer cell proliferation, highlighting PKM1 as a potential therapeutic target for this type of neoplasia [23]. In this regard, the PKM1/PKM2 ratio was found to be higher in SCCOPT, in PM and in NC-OV, whereas PKM2 was present only in HGSC (Figure 3B). PKM1 expression in SCCOPT and in PM suggests that at least this tumor case may predominantly use glycolytic metabolic routes to sustain proliferation and growth, and, hence, may be similar to SCLC in terms of energetic profile. To further test for the determinants of the Warburg signature, the expression of some glycolytic markers, such as LDHA and GAPDH, was compared with the amount of oxidative phosphorylation (OXPHOS) enzymes. While glycolytic proteins were found to be upregulated in the PM sample (Figure 3C), mitochondrial oxidative proteins showed a marked reduction in both cancer tissues, suggesting that the latter had a low-OXPHOS signature and largely relies on glycolysis [24] (Figure 3D). This was further corroborated by the high expression level of glucose transporter GLUT-1 compared to the adjacent non-cancer tissue, as displayed by immunohistochemistry analysis (Figure 3E). Finally, we reasoned that a potential explanation for the glycolytic signature may lie within the mitochondrial DNA (mtDNA) [25]. Since mtDNA mutations are reported as a balance needle in determining the metabolic signature of cancers [26,27], but not in these tumor types, we sequenced the whole mtDNA and detected the presence of the m.8828A > G missense mutation of a conserved residue (p.N101S) in the MT-ATP6 gene encoding the corresponding mitochondrial complex V (CV, i.e., ATP synthase) subunit (Figure 3F); such a finding has never been reported regarding somatic cancer tissues before. The mutation was present only in the SCCOPT and PM but not in the non-cancer tissue, and was therefore deemed to be a somatic, tumor-specific mutation. The variant was nearly homoplasmic, i.e., was present at a very high load in cancer cells, and was predicted to be pathogenic [28]. This mutation was already reported in a patient with Charcot-Marie-Tooth type 2, but its potential pathogenetic nature remains uncertain [29]. The missense mutation did not seem to cause an impairment in CV assembly, meaning that the CV was equally assembled in the three samples analyzed (Figure 3G). However, these findings did not allow us to exclude the possibility that such a pathogenic mutation may have triggered an impairment in the activity of CV and, in turn, affected the energetic status, likely favoring a glycolytic profile.

## 3. Discussion

Small cell neuroendocrine carcinoma (SmCC) is most frequently found in the lung (95%), but extra-pulmonary SmCC has been reported in almost every organ, even in the ovary [30]. SCCOPT is a challenging diagnosis, as it is not easily distinguished pre-operatively from common epithelial ovarian cancers (i.e., HGSC), and differential diagnoses will include metastatic neuroendocrine carcinoma, germ cell and granulosa/sex-cord tumors. Despite its rarity, correct diagnoses could result in a better clinical, prognostic and predictive definition of this type of cancer. Conventional treatment of SCCOPT is based on surgery followed by chemotherapy. As with other extra-pulmonary SmCCs, the chemotherapy regimen for SCCOPT is similar to those for SCLC [5]. In this study we obtained the miRNA expression profile of a SCCOPT case, confirming a miRNA expression pattern resembling those of both neuroendocrine tumors and SCLCs. We then applied a recently developed molecular tool, designed to identify the primary site of tumor samples, to assess the similarity between the SCCOPT and 17 different tumor types. Using this approach, we confirmed the highest level of similarity with SCLC. As previously described, the mTOR signaling axis is commonly upregulated in SCLC, and its inhibition has been reported to prevent cell growth and increase patient survival [31]. However, clinical studies on relapsed SCLC using the Everolimus mTOR inhibitor showed limited antitumor activity when used as single therapeutic agent [32]. Nonetheless, several studies have reported improved anticancer activity of Everolimus when combined with other chemotherapeutics [17,33,34]. In this regard, the hyperactivation of mTOR-downstream S6K that we found in this SCCOPT case indicates the similarity of this tumor to SCLC, suggesting that it was potentially targetable with mTOR inhibitors. Moreover, it is known that mTOR signaling may be crucial in defining the metabolic properties of cancer cells. In this context, SCLC has been reported as a high-glycolytic type of tumor, since it displays overexpression of the common metabolic markers of glycolysis (i.e., PKM1 and LDHA) [35,36,37]. In agreement with these findings, the glycolytic signature that we observed in our SCCOPT case and its peritoneal metastasis highlights the similarity of this case with SCLC. Furthermore, the occurrence of a unique damaging mutation in the mitochondrial ATP synthase, which is downstream of functional respiratory complexes, may suggest the need for a compensatory glycolytic flux. Alternatively, the mutation may have been favorably selected, since oxidative metabolism is not preferentially used here, likely due to deregulation of the mTOR pathway, and therefore, to the absence of selective pressure on the respiratory chain enzymes. 

## 4. Conclusions

We analyzed a rare neoplasm that was diagnosed as SCCOPT. MicroRNA profiling indicated a pronounced similarity with neuroendocrine tumors and lung cancer, rather than HGSC. Furthermore, mTOR-downstream activation and a glycolytic signature suggested additional similarities with SCLC. These findings are relevant in the field of rare cancer entities, such as the SCCOPT we describe herein, whose deeper molecular categorization may prove to be of value in selecting the correct treatment. Since the use of molecular biology tools is becoming routine, we attempted to demonstrate that genetics and molecular and metabolic parameters represent a vast milieu of markers for the identification of similarities, pathways and profiles that may, in the near future, orient personalized choices regarding therapeutic regimens.

## 5. Materials and Methods

**Tissue samples collection.** The patient was enrolled in the MiPEO (Mitochondria in Progression of Endometrial and Ovarian Cancer) study, approved by the local ethical committee at S. Orsola Hospital, Bologna. Primary SCCOPT and peritoneal metastasis tissues were collected from surgical specimens after histopathological analysis. Tissues were cut into pieces, snap frozen and stored at −80 °C or formalin-fixed and paraffin-embedded. Non-cancer and common ovarian high-grade serous cancer tissues were used as controls. Hematoxylin and eosin sections were reviewed to identify paraffin blocks with tumor areas.

**Immunohistochemistry.** Analyses were performed on formalin-fixed and paraffin-embedded tissue sections using the following specific antibodies: TTF-1 (clone 8G7G3; mouse monoclonal antibody; ready to use, Ventana Medical System, Inc., Monza, Italy), p53 (clone DO-7; mouse monoclonal antibody; ready to use, Ventana Medical System, Inc.), chromogranin A (clone LK2H10; mouse monoclonal antibody; ready to use; Ventana Medical System, Inc.), and GLUT-1 (clone SP168; rabbit monoclonal primary antibody, ready to use; Ventana Medical System, Inc.). All sections were immunostained with automatic immunostaining Benchmark Ultra-Roche Diagnostics. 

**MicroRNA profiling analysis.** RNA was extracted from SCOOPT tumor formalin-fixed paraffin embedded (FFPE) tissue slices using an miRNeasy FFPE kit (Qiagen, Milan, Italy) and reverse transcribed using miRCURY LNA RT Kit (Qiagen). The SCOOPT tumor-specific 81 miRNA signature was obtained by droplet digital PCR (Bio-Rad, Milan, Italy) using an Evagreen-based protocol [38] and normalized on *SNORD44* reference small RNA. The miRNA signature was used to perform a similarity and predictive analysis against a panel of 55 primary tumor samples belonging to 17 tumor classes that were published by Laprovitera et al. [38]. Prediction of the tissue-of-origin was performed as previously described [11]. The clustering analysis based on Pearson correlation was performed using GeneSpring GX software v.14.9.1 (Agilent Technologies, Milan, Italy).

**Crude mitochondria preparation**. Crude mitochondria from snap-frozen tissues were obtained by shredding in ice-cold Sucrose-Mannitol Buffer (200 mM mannitol, 70 mM sucrose, 1 mM EGTA and 10 mM Tris-HCl at pH 7.6) and homogenizing using a glass/Teflon Potter homogenizer. The obtained samples were centrifuged at 600 g for 10 min at 4 °C. The resulting supernatant was centrifuged at 10,000 g for 20 min at 4 °C to separate crude mitochondria from the remaining sub-cellular fractions of the sample. Crude mitochondria were used for SDS–PAGE and Blue Native-PAGE experiments.

**SDS-PAGE and Western blotting.** Whole lysates and crude mitochondria from tissue samples were prepared in RIPA buffer (50 mM Tris–HCl pH 7.4, 150 mM NaCl, 1% SDS, 1% Triton, 1 mM EDTA, pH 7.6) supplemented with inhibitors of proteases (Thermo Scientific #A32955) and phosphatases (Thermo Scientific, Milan, Italy, #A32957) and quantified using Bradford’s method [39]. Whole lysates and crude mitochondria were separated by SDS-PAGE using a TGX Fast-Cast-TM Acrylamide kit (Bio-Rad #1610173). Membranes were blocked at 37 °C for 30 min in TBS-Tween/BSA 5% and incubated with primary antibodies using the following conditions and dilutions: anti-S6K (Cell Signaling Technology, Milan, Italy, #2708S) 1:1000 overnight at 4 °C, anti-phospho-S6K (Thr389) (Cell Signaling Technology, #9206S) 1:1000 overnight at 4 °C, anti-vinculin (GeneTex, Milan, Italy, #GTX113294) 1:2000 1h at RT, anti-PKM1 (Proteintech, Milan, Italy, #15821-1) 1:1000 overnight at 4 °C, anti-PKM2 (Proteintech, #15822-1) 1:1000 overnight at 4 °C, anti-LDHA (Sigma-Aldrich, Milan, Italy, #SAB2108638) 1:1000 overnight at 4 °C, anti-GAPDH (Sigma-Aldrich, #G8795) 1:20,000 1 h at RT, and anti-total OXPHOS (Abcam, Cambridge, UK, ab110411) 1:1000 overnight at 4 °C, anti-VDAC (Abcam, #ab154856) 1:2000 1 h at RT. Membranes were washed using TBS-Tween (0.05% Tween (Sigma-Aldrich, #P9416) in Tris-buffered saline). Secondary antibodies (Jackson Immuno-Research Laboratories, Milan, Italy, #111035144 and #111035146) were incubated for 1 h at RT using 1:5000 dilutions in TBS-Tween. Membranes were developed using Clarity Western ECL Substrate (Bio-Rad, Milan, Italy, #1705061), and detection was performed with ChemiDoc (Bio-Rad). A densitometric analysis was performed by using ImageJ (v.1.52t, National Institute of Health, Bethesda, MD, USA) [40].

**Blue Native-PAGE and Western blotting.** Blue Native-PAGE (BN-PAGE) was performed on crude mitochondria fractions from tissue samples as previously described [41]. Crude mitochondria were solubilized in 1.5 M aminocaproic acid and 50 mM Bis-Tris/HCl at pH 7 with the addition of 4μg digitonin/μg total proteins and incubated at 4 °C for 5 min [42]. Proteins (70 μg) were loaded on 3–12% native PAGE gradient gel and separated at 150 V (4 °C for 3 h). For separation, cathode buffer A (50 mM tricine, 7.5 mM imidazole, 0.002% Coomassie Brilliant Blue G250, pH 7), cathode buffer B (50 mM tricine, 7.5 mM imidazole, pH 7), and anode buffer (25 mM imidazole, pH 7) were used. Cathode A was replaced with cathode B when the frontline was halfway from the gel. Samples were then transferred onto polyvinylidene fluoride membranes using a Turbo-pack system (Bio-Rad #1704159SP5). Membranes were blocked at 37 °C for 1 h and incubated with primary antibodies using the following conditions and dilutions: anti-ATP5A (CV) (Abcam, # ab14748) 1:1000 overnight at 4 °C, anti-SDHA (CII) (Thermo Scientific, #459200) 1:10,000 2 h at RT. Secondary antibodies (Jackson Immuno-Research Laboratories, #111035144 and #111035146) were incubated for 1 h at RT using 1:5000 dilutions in TBS-Tween. Membranes were developed using Clarity Western ECL Substrate (Bio-Rad #1705061), and detection was performed with ChemiDoc (Bio-Rad).

**MtDNA sequencing and variants prioritization**. Total DNA was extracted using a Mammalian Genomic DNA Miniprep Kit (Sigma-Aldrich, #G1N350), according to the manufacturer’s protocol. PCR reactions were performed using a KAPA2G Fast PCR Kit (Sigma Aldrich) with a set of 46 primer pairs, as previously described [43]. The 46 purified PCR products were used for direct sequencing with a BigDye kit version 1.1 (Thermo Fisher Scientific), and the sequences were run in an ABI 3730 Genetic Analyzer (Thermo Fisher Scientific). To perform the analysis, electropherograms were aligned with the rCRS mitochondrial reference sequence using SeqScape version 2.5 software (Applied Biosystems, Waltham, MA, USA). To annotate and prioritize the mitochondrial variants, FASTA files from primary cancer and peritoneal metastasis (GenBank Accession Numbers OP342772-OP342773) were used as the input for MToolBox [44]. Selected variants were analyzed using HmtVar (https://www.hmtvar.uniba.it, accessed on 23 June 2022) to infer pathogenicity [45]. Information from previous reports about the observed variant was retrieved from the following human mitochondrial databases (last accessed on 13 September 2022): MITOMAP: A Human Mitochondrial Genome Database (http://www.mitomap.org), ClinVar (https://www.ncbi.nlm.nih.gov/clinvar/) [46], MseqDR (https://mseqdr.org/) [47] and HmtDB (https://www.hmtdb.uniba.it/) [48]. The following datasets were also consulted: The Cancer Mitochondrial Atlas (TCMA) (http://ibl.mdanderson.org/tcma/, accessed on 28 July 2022) [49] and the catalog of mtDNA somatic mutations in Yu et al. [50]. MtDNA mutations were confirmed using a second PCR reaction; the sequencing of matched non-tumor tissue allowed us to assess their tumor-specific origin.

**Next-generation sequencing of neoplastic lesions.** To characterize the tumor mutation profile, primary cancer, peritoneal metastasis and non-tumor tissue were analyzed using a laboratory-developed panel including 22 oncogenes and tumor suppressor genes (human reference sequence hg19/GRCh37), as previously described [51]. Briefly, about 30 ng of input DNA was used to prepare the NGS libraries, and templates were sequenced using a MiSeq sequencing platform (Illumina Inc., San Diego, CA, USA), according to the manufacturer’s instructions. The obtained sequences were analyzed using VariantStudio Software (Illumina Inc., San Diego, CA, USA) and IGV software (Integrative Genome Viewer, v.2.12.2—https://software.broadinstitute.org/software/igv/, accessed on 28 July 2022, University of California, CA, USA). Only variants present in at least 5% of the total number of reads analyzed and observed in both strands were considered.

## Figures and Tables

**Figure 1 ncrna-08-00064-f001:**
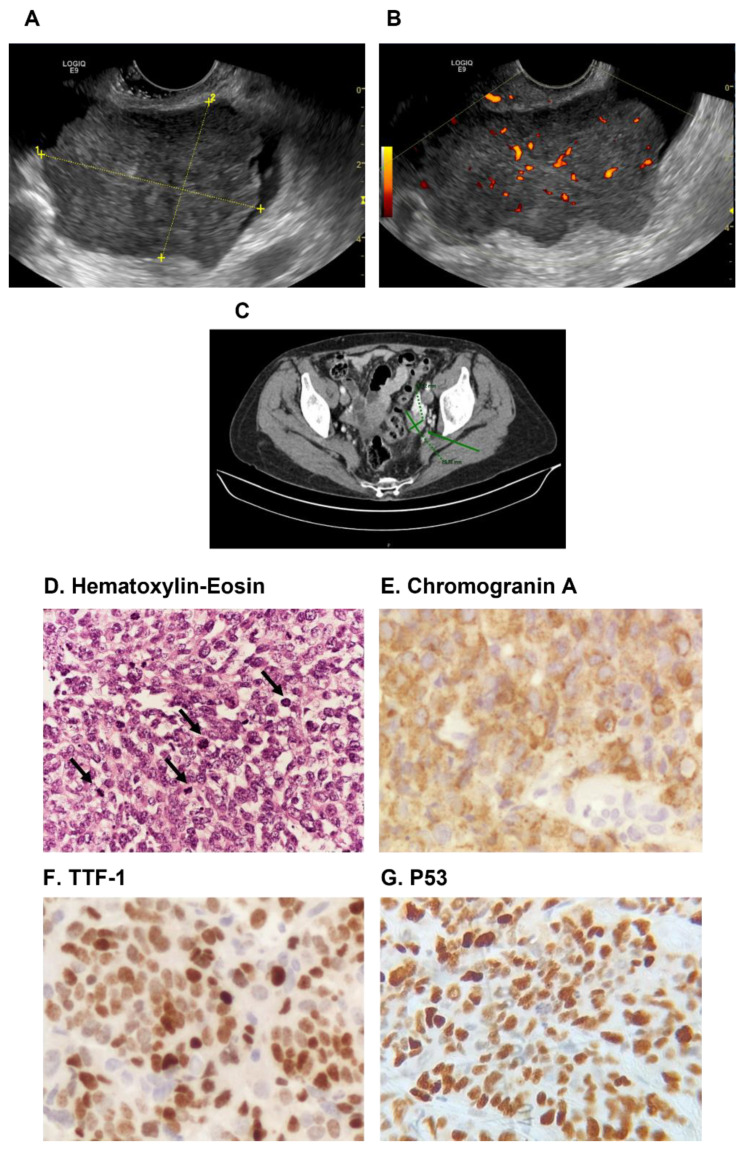
(**A**) Transvaginal ultrasound showing the left adnexal solid mass (7 cm), and (**B**) blood flow (CS 3) in the pelvic mass. (**C**) Abdominal CT scan showing the left ovarian mass after chemotherapy (green arrow). (**D**) Hematoxylin-Eosin staining showing tumor cells with scant cytoplasm and stippled chromatin (original magnification, 400×); black arrows indicate atypical mitotic figures. (**E**) Immunohistochemical positivity for chromogranin A, and (**F**) TTF-1 in SCCOPT (original magnification, 400×). (**G**) Immunohistochemical overexpression of p53 in SCCOPT (abnormal/mutation-type pattern; original magnification, 400×).

**Figure 2 ncrna-08-00064-f002:**
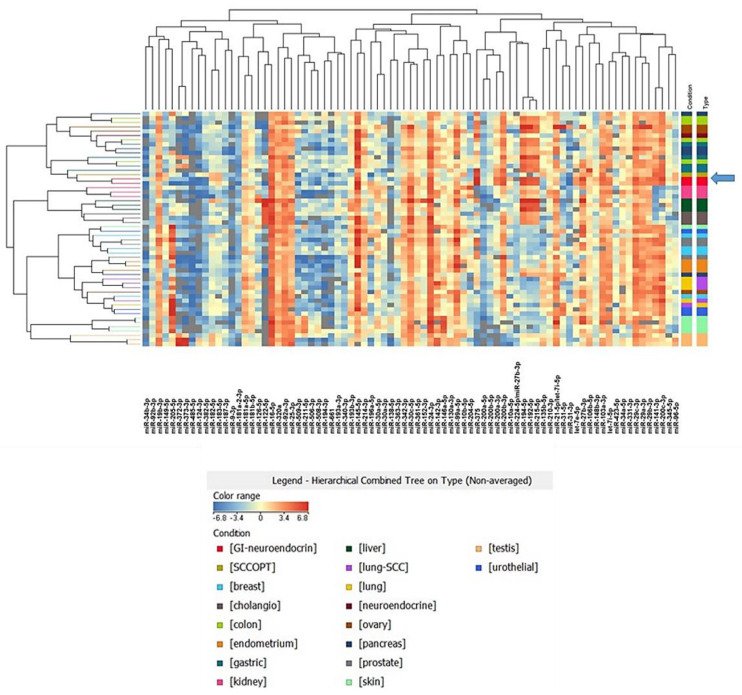
Cluster analysis of SCCOPT (indicated by the arrow) and 79 samples from 17 different cancer types based on the expression of tumor-specific miRNAs detected with droplet digital PCR. Pearson correlation was used as a measure of similarity among samples.

**Figure 3 ncrna-08-00064-f003:**
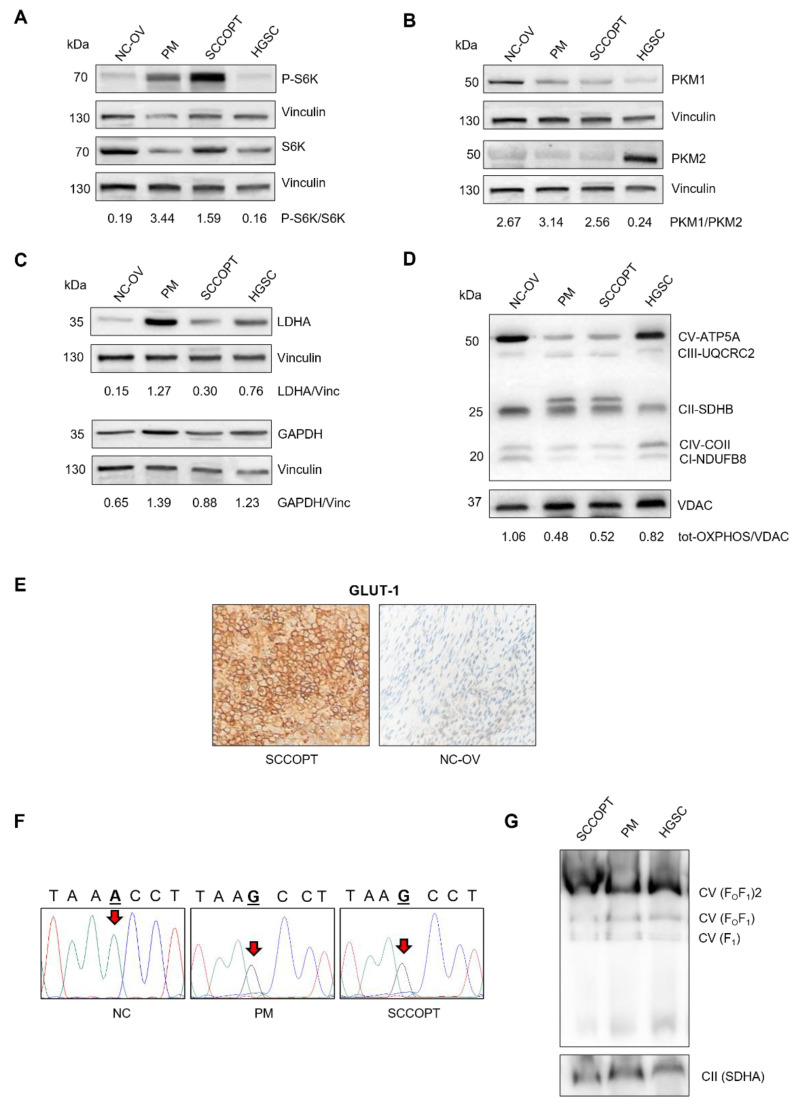
(**A**) Representative western blot analysis of total S6K (S6K) and phosphorylated S6K (P-S6K) in total lysates from non-cancer ovarian tissue (NC-OV), peritoneal metastasis (PM), SCCOPT and ovarian high-grade serous carcinoma (HGSC). Vinculin (Vinc) was used as a loading control. Densitometry values are shown as P-S6K/S6K ratio. (**B**,**C**) Representative western blot analysis of PKM1 and PKM2 (**B**) and LDHA and GAPDH (**C**) expression levels in total lysates from NC-OV, PM, SCCOPT and HGSC tissues. Vinculin was used as a loading control. Densitometry values are shown as the following ratios: PKM1/PKM2 (**B**), LDHA/Vinc and GAPDH/Vinc (**C**). (**D**) Representative western blot analysis of OXPHOS enzymes (CV-ATP5A, CIII-UQCR2, CII-SDHB, CIV-COII, CI-NDUFB8) in crude mitochondria obtained from NC-OV, PM, SCCOPT and HGSC tissues. VDAC was used as loading control. Densitometry values are shown as tot-OXPHOS/VDAC (tot-OXPHOS: the sum of densitometry values for each lane). (**E**) Immunohistochemical staining of GLUT-1 in SCCOPT and in the residual ovarian parenchyma (NC-OV). Magnification: 400×. (**F**) Sequence analysis of mtDNA variants in non-cancer (NC), SCCOPT and PM tissues. Red arrows indicate the mutated bases. (**G**) Representative western blot analysis of F_o_F_1_ ATPase (CV) species in crude mitochondria isolated from SCCOPT, PM and HGSC tissues. The assay was performed using a BN-PAGE/Western blot assay, as reported in Materials and Methods. SDHA (CII) was used as a loading control.

**Table 1 ncrna-08-00064-t001:** Most expressed MiRNAs in the SCCOPT sample. MiRNA copies were quantified using droplet digital PCR and normalized on *SNORD44* expression.

MiRNA	Normalized Expression
miR-375	16.68
miR-141-3p	10.05
miR-200c-3p	9.34
miR-16-5p	9.31
miR-103a-3p	6.72
miR-200b-3p	6.02
miR-19b-3p	5.99
miR-24-3p	5.95
miR-145-5p	4.69
miR-92a-3p	4.58
miR-27b-3p	2.48
miR-30c-5p	2.40
miR-106b-5p	2.29
miR-200a-3p	2.24
miR-10b-5p	2.20

## Data Availability

Sequencing data are available in NCBI—Sequence Read Archive (SRA) (PRJNA874675). Genome vcf (gvcf) files have been provided in the Appendix A. Mitochondrial sequences of SCCOPT and PM samples were deposited in the public database (GenBank Accession Numbers OP342772-OP342773).

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
