# Peer review of "MicroRNA and Metabolic Profiling of a Primary Ovarian Neuroendocrine Carcinoma Pulmonary-Type Reveals a High Degree of Similarity with Small Cell Lung Cancer"

_ncrna, 2022, doi:10.3390/ncrna8050064_

Round 1
Author Response
- The ovarian neuroendocrine carcinomas or small cell carcinoma of ovarian (SCCO) are described
in literature as two types – the hypercalcemic type (SCCOHT) and pulmonary type (SCCOPT) [Yin et. al]. From the description of the case, it is suggestive of SCCOPT, if true, it is requested to retain one annotation to define this carcinoma instead of a new term “OPSmCC”. Having said that, the implication of the role of mutations indicated and genetic alterations of TP53 at Line 71,72 in the introduction and Line 119, supporting the mutation of TP53 c.488A>G in results both belong to SCCOHT type. How do authors justify this misleading citation? A thorough rationale need to support the identifications and/or citation. The read depths for this A>G SNV across primary cancer, peritoneal metastasis, and non-tumor tissues, should be mentioned in the main text and should include reference and alternate allele counts.
Yin et. al. (2018). Primary ovarian small cell carcinoma of pulmonary type with coexisting endometrial carcinoma in a breast cancer patient receiving tamoxifen. Medicine (Baltimore). 97, 23. DOI: 10.1097/MD.0000000000010900
According to the recent 2020 WHO classification of female tumors, small cell carcinoma of the ovary of the hypercalcemic type is an undifferentiated tumor often associated with hypercalcemia, unrelated to small cell neuroendocrine carcinoma (pulmonary) that arises almost exclusively in young women. In the case presented the clinical features (elderly patient, no evidence of hypercalcemia) and immunohistochemical profile are diagnostic for SCOOPT (positivity for TTF1, diffuse neuroendocrine markers, unlike SCCOHT which is TTF1 negative with variable expression of neuroendocrine markers). TP53, as it is known, is one of the most common tumor suppressors mutated in neuroendocrine carcinomas. We thank the review for this annotation, we have changed the acronym OPSmCC to SCOOPT, added the suggested citation and the following sentence “Ovarian neuroendocrine carcinomas or small cell carcinoma of the ovary are described in literature as two types – the hypercalcemic type and pulmonary type [4]”.
The coverage (total reads) for each analyzed specimen and the VAF (variant allele frequency) were the following: primary tumor 1726x (TP53 c.488A>G VAF: 92%); peritoneal metastasis 1141x (TP53 c.488A>G VAF: 82.7%), non-neoplastic tissue 4774x (no TP53 c.488A>G variant detected).
- Line 135: miR-34b, is reported to be downregulated in SCLC cell lines (Mizuno et al.) what are the expression levels in the present study? In another study cited by the authors report miR-34a has higher target genes compared to miR-34b and miR-34c (Navarro et al.). The higher levels of miR-34a is shown to be both positive and negative regulators of TP53 gene, playing the role of tumor suppressor and promoting cell death. However, miR-34a is not associated with TP53 mutations as stated in Line 136, 137. Further, miR-34 is not transcribed by P53, it is transcriptionally activated. If my understanding is correct P53 is not the host gene for miR-34. This is misleading or confusing, and therefore need correction or elaborate on the statements in the main text.
Mizuno et al. (2017). The microRNA expression signature of small cell lung cancer: tumor suppressors of miR-27a-5p and miR-34b-3p and their targeted oncogenes. Journal of Human Genetics (2017) 62, 671–678. Navarro et al. (2015). miR-34 and p53: New Insights into a Complex Functional Relationship. PLoS One. 10, 7.
We provided all the miRNA levels (copies) detected by droplet digital PCR in the SCCOPT sample, normalized on SNORD44 reference gene, as a supplementary table 1.
Summarizing the relationship between miR-34 and P53 is not an easy task. Their regulation goes in two ways: TP53 (WT) is a transcription factor for miR-34 miRNA family (including miR-34 a, b, c) and miR-34 has TP53 among its target genes. In the presence of mutations in TP53 gene, the miR-34 family is therefore less transcribed. And this is the reasoning behind the sentence written in the manuscript. Below is the summary of the main literature supporting the physiological interaction between P53 and miR-34a.
- Bommer GT, Gerin I, Feng Y, et al. p53-mediated activation of miRNA34 candidate tumor-suppressor genes. Current biology : CB 2007; 17(15): 1298-307.
- Chang TC, Wentzel EA, Kent OA, et al. Transactivation of miR-34a by p53 broadly influences gene expression and promotes apoptosis. Molecular cell 2007; 26(5): 745-52.
- He L, He X, Lim LP, et al. A microRNA component of the p53 tumour suppressor network. Nature 2007; 447(7148): 1130.
- He L, He X, Lowe SW, Hannon GJ. microRNAs join the p53 network--another piece in the tumour-suppression puzzle. Nature reviews Cancer 2007; 7(11): 819-22.
- Raver-Shapira N, Marciano E, Meiri E, et al. Transcriptional activation of miR-34a contributes to p53-mediated apoptosis. Molecular cell 2007; 26(5): 731-43.
- Tarasov V, Jung P, Verdoodt B, et al. Differential regulation of microRNAs by p53 revealed by massively parallel sequencing: miR-34a is a p53 target that induces apoptosis and G1-arrest. Cell cycle (Georgetown, Tex) 2007; 6(13): 1586-93.
- Georges SA, Biery MC, Kim SY, et al. Coordinated regulation of cell cycle transcripts by p53-Inducible microRNAs, miR-192 and miR-215. Cancer research 2008; 68(24): 10105-12.
- Yamakuchi M, Lowenstein CJ. MiR-34, SIRT1 and p53: the feedback loop. Cell cycle (Georgetown, Tex) 2009; 8(5): 712-5.
- Asslaber D, Piñón JD, Seyfried I, et al. microRNA-34a expression correlates with MDM2 SNP309 polymorphism and treatment-free survival in chronic lymphocytic leukemia. Blood 2010; 115(21): 4191-7.
- Bhatt K, Zhou L, Mi QS, Huang S, She JX, Dong Z. MicroRNA-34a is induced via p53 during cisplatin nephrotoxicity and contributes to cell survival. Molecular medicine (Cambridge, Mass) 2010; 16(9-10): 409-16.
- Cannell IG, Kong YW, Johnston SJ, et al. p38 MAPK/MK2-mediated induction of miR-34c following DNA damage prevents Myc-dependent DNA replication. Proceedings of the National Academy of Sciences of the United States of America 2010; 107(12): 5375-80.
- Nalls D, Tang SN, Rodova M, Srivastava RK, Shankar S. Targeting epigenetic regulation of miR-34a for treatment of pancreatic cancer by inhibition of pancreatic cancer stem cells. PloS one 2011.
- Wang Y, Li X, Hu H. Transcriptional regulation of co-expressed microRNA target genes. Genomics 2011; 98(6): 445-52.
- Concepcion CP, Han YC, Mu P, et al. Intact p53-dependent responses in miR-34-deficient mice. PLoS genetics 2012.
- Tan J, Fan L, Mao JJ, et al. Restoration of miR-34a in p53 deficient cells unexpectedly promotes the cell survival by increasing NFκB activity. Journal of cellular biochemistry 2012; 113(9): 2903-8.
- Bisio A, De Sanctis V, Del Vescovo V, et al. Identification of new p53 target microRNAs by bioinformatics and functional analysis. BMC cancer 2013.
- Kim NH, Cha YH, Kang SE, et al. p53 regulates nuclear GSK-3 levels through miR-34-mediated Axin2 suppression in colorectal cancer cells. Cell cycle (Georgetown, Tex) 2013.
- Tay Y, Tan SM, Karreth FA, Lieberman J, Pandolfi PP. Characterization of Dual PTEN and p53-Targeting MicroRNAs Identifies MicroRNA-638/Dnm2 as a Two-Hit Oncogenic Locus. Cell reports 2014.
- Xiao Z, Li CH, Chan SL, et al. A small molecule modulator of the tumor suppressor miRNA-34a inhibits the growth of hepatocellular carcinoma. Cancer research 2014.
- Wellenstein MD, Coffelt SB, Duits DEM, et al. Loss of p53 triggers WNT-dependent systemic inflammation to drive breast cancer metastasis. Nature 2019; 572(7770): 538-42.
- Amit M, Takahashi H, Dragomir MP, et al. Loss of p53 drives neuron reprogramming in head and neck cancer. Nature 2020; 578(7795): 449-54.
- Fawzy MS, Ibrahiem AT, AlSel BTA, Alghamdi SA, Toraih EA. Analysis of microRNA-34a expression profile and rs2666433 variant in colorectal cancer: a pilot study. Scientific reports 2020.
- Kang M, Tang B, Li J, et al. Identification of miPEP133 as a novel tumor-suppressor microprotein encoded by miR-34a pri-miRNA. Molecular cancer 2020.
Figure 2A. The figure is important in classifying the OPSmCC across other tumor types. However, (although common in literature), this color combination of red/green heatmap can be challenging to people with red/green color-blindness. Thus, the authors are requested to change the colors of heatmap to be more easily visualized for example, cyan-Black-Red, Green-Black- Magenta, purple-white-orange, light blue-black-yellow etc. Further, an arrow pointing at the OPSmCC type will ease the understanding to many readers.
The Figure has been changed accordingly.
Figure 3G. The peaks surrounding m.8828 A>G also seem to be low compared to NC-OV, and the figure represent R instead of G, does that mean it is heterogenous SNV with A or G? What additional evidence can prove that this is actual variation and not a readout error? The FASTA file should be provided as supplementary files or submitted to NCBI GenBank and provide an accession number in the “Data Availability Statement”.
With the aim to ease these data interpretation, we have now indicated in the figure the mutant peak as G, rather than using the IUPAC nomenclature R (G/A). We would like to point out that the very low peak the reviewer refers to would be A, the wild-type allele, not the mutant G, which is the highest peak. Indeed, we hope that indicating the mutation as nearly homoplasmic in the text did not generate confusion as to the fact that the mutation load is very high, and therefore not a readout error. The fact that the same somatic mutation is detected in two separate samples (metastasis and the primary tumor), with two separate DNA extractions and PCRs also contributes to rule out this as a readout error.
The indication of near homoplasmy ought to be given because we are not able to rule out that either a very low percentage of the wild type A allele is indeed present in cancer cells, or it is given by a low contamination of the tumor tissue by non-cancer cells.
We now provide, according to the reviewer’s request, the FASTA file of the mitochondrial sequences of SCCOPT and PM in the public database (GenBank Accession Numbers OP342772-OP342773).
Line 343: Needs clarification on the type of sequencing performed, is it whole genome or exome sequencing or targeted gene sequencing, and this should be mentioned throughout the main text. Further, the sequencing data (e.g.: FASTQ), Binary Alignment Map, (BAM) files should be submitted to NCBI – Sequence Read Archive (SRA) and provide the accession in the main text under “Data Availability Statement”. Also, the variant calling files generated during the analysis and associated results should be provided as the supplementary files.
The NGS analysis was performed using a targeted laboratory-developed multi-gene panel as reported inMaterials and Methods section and throughout the manuscript, as requested. Sequencing data have been submitted to NCBI – Sequence Read Archive (SRA) (SUB11979346, Project: PRJNA874675). Genome vcf (gvcf) files have been provided in the supplementary files.
The results of microRNA expression profiling should be provided as supplementary file.
The 81-miRNA expression in the SCOOP sample is now provided as supplementary file.
Minor
1. Line 73: The authors are requested to replace the use of “exploit miRNA” to “profiled select 81 miRNAs”. Since there are ~1,900 miRNAs reported in miRBase alone and exploring only 81 miRNAs can’t be considered to “exploit miRNA”. microRNAs have unique cell specific expression and it may have different expression settings across tumor origin. With the rare tumor type collection such as this (SCOOPT), one would recommend exploring all small RNAs by NGS-based small-RNA sequencing and finding functional target genes associated with this cancer.
We applied a predictive test for tumor primary identification that was previously established in Ferracin’s lab (PMID: 34075699 and PMID: 21630269) to this rare SCOOPT sample. Here we present the outcome of this molecular test. We provided further clarifications in the manuscript.
We agree with the reviewer that a small RNA sequencing of this SCOOPT sample could provide useful information, but it would not have changed the outcome of the predictive test.
Resolution of Figure 2A can be improved.
Figure 2A resolution has been improved and named as Figure 2.
- Line 272: Moreover, a deeper molecular and genetic characterization of the tumor, such as the one provided by miRNA profiling, can constitute an increasingly cheap and feasible strategy to provide a personalized choice of the therapeutic regimen. The discussion or use of a specific therapeutics with respect to miRNA profiling or other molecular identifications in this study does not direct towards a suitable personalized therapy. Can the authors describe the role of this study to help/support in therapeutic choice?
We have now restructured the conclusions to make them more adherent to the data reported and to the aim of the study, upon request by other reviewers as well. We apologize to the reviewer for overstating that indeed, from this single case, miRNA profiling may directly orient per se the therapeutic regimen. With respect to how molecular findings may direct towards personalized therapeutic choices, instead, the reviewer may refer to the part of the discussion concerning the data on mTOR and its effectors we reported, and the use of mTOR inhibitors, such as the Everolimus we briefly mention.
Reviewer 2 Report
The general idea of the paper is good, however, it is not well presented. The description of the results is confusing and it is hard to follow. The discussion is not well structured and conclusion is missing.
Questions and recommendations.
1.- The authors mention “The right ovary and perimetrium presented hypoechoic nodules. Pelvic free fluid was present and Douglas peritoneum was thickened. Trans-abdominal ultrasonography evidenced a 4 cm para-aortic metastatic lymph node. Computed tomography (CT) scan confirmed the sonographic findings. The patient, due to the poor general condition, was unfit for surgery, therefore a transvaginal ultrasound-guided core needle biopsy of the left adnexal mass was performed. The pathological examination showed a high-grade carcinoma compatible with a gynecological origin.”
However, the evidence was not shown, therefore, I recommend to add CT before and after to compare.
2.- The author mention “Mitotic figures were numerous, some with atypical features.”
Can you show this features in the figure with an arrow.
3.- The sentence is redacted as if it had been done in the present work “Immunohistochemical staining showed negativity for WT-1, PAX-8 and estrogen receptor/progesterone receptor, excluding a tubo-ovarian high-grade serous carcinoma (HGSC) [3,8]”
This was not evaluated in the author’s work.
4.- The authors mention “Synaptophysin and chromogranin were diffusely expressed (Fig. 1E), suggesting a neuroendocrine differentiation [9].” However, Synaptophysin was not evaluated in this work.
5.- The authors mention “The tumor cells were also positive for pan-cytokeratin AE1/AE3, and TTF-1 (Fig. 1F), p53 (Fig. 1G).” But pan-cytokeratin AE1/AE3 was not evaluated in this work.
6.- The authors mention “These morphological and immunohistochemical findings were similar to those of SCLC, but a metastatic origin was excluded by the CT scan.”
Can you explain how a CT determine metastatic origin of a tissue? It is the first time that I read this asseveration.
7.- The authors mention “The analysis allowed to detect the TP53 p.Y163C (c.488A>G, exon 5) missense mutation in all cancer specimens, correlated to a strong and diffuse p53 overexpression (Fig. 1G), confirming the already reported frequent alteration of TP53 in OPSmCCs.”
Can you explain how a missense mutation correlates with protein expression? Does the p53 antibody recognize the amino acid change? Did you have antibodies that recognize p53 with the amino acid C and Y? The confirmation of TP53 alteration by the mutation could be suggested but it was not demonstrated in the present work.
8.- The authors mention “We performed a clustering analysis, using Pearson correlation as similarity measure, unveiling that “neuroendocrine” tumors have the most similar expression profile (Fig. 2A).” Can you describe and/or mention which tumors are those?
9.- The authors mention “Among the miRNAs most expressed in OPSmCC, we detected miR-375, miR-141-3p, miR-200c-3p, miR-16-5p and miR-103a-3p (Fig. 2B). Did you compare the expression with a normal tissue to corroborate their increased expression?
10.- The authors mention “MiR-141-3p and miR-200 family are up- 141
regulated in prostate cancers and regulate the metastatic process in many cancer types. In ovarian cancer the overexpression of miR-200 family has been associated with a mesothelial-to-epithelial transition and a more aggressive phenotype [14]. This evidence confirmed the similarity of OPSmCC to lung neuroendocrine carcinomas, despite the different site of origin.”
Is the expression of miR-141-3p, miR-200 family are only up-regulated in neuroendrocrine carcinomas to confirm the similarity of OPSmCC to lung neuroendocrine carcinomas?
11.- The authors mention “PKM1 expression in OPSmCC and in the metastasis confirmed the similarity of this tumor with SCLC, suggesting that this tumor case may use predominantly glycolytic metabolic routes to sustain proliferation and growth.”
The samples used are from ovary, therefore, you cannot confirm that they are similar to SCLC. Additionally, samples from peri-metastatic, OPSmCC and HGSC are similar and you pull apart or discuss only peri-metastatic and OPSmCC, therefore describing an unexisting mechanism regarding PKM1 protein.
12.- The authors mention “While glycolytic proteins were found up-regulated in the peritoneal metastasis (Fig. 3C), mitochondria oxidative proteins showed a marked reduction in both cancer tissues, suggesting the latter to have a low-OXPHOS signature and to rely on glycolysis [24] (Fig. 3D).” Except for CII-SDHB that is overexpressed in cancer tissues. You can argue and discuss why.
13.- The authors mention “This was confirmed also by the high expression level of glucose transporter GLUT-1, as displayed by immunohistochemistry (Fig. 3E).”
You cannot confirm a low-OXPHOS signature in cancer tissue based in GLUT-1 expression showing only a photograph of OPSmCC.
14.- The authors mention “Intriguingly, respiratory complexes I (CI) and III (CIII) activity was higher in OPSmCC and in the metastasis compared to non-tumor tissue, revealing that oxidative metabolic routes may be functional in the analyzed case (Fig. 3F).”
Can you explain why in non-tumor tissue the CI does not have activity?
In this sentence the activity between OPSmCC, peri-metas and HGSC are similar, you are making a bias in the description of your results.
15.- The authors mention “Furthermore, the mutation was already reported in a patient with Charcot-Marie-Tooth but his potential pathogenetic nature remains uncertain [29]. Based on MT-ATP6 immunohistochemistry (Fig. 3H), the missense mutation did not seem to hamper the protein amount and the assembly of CV.”
How is the expression in the other tissues? You didn´t include other tissues in your study. Can you argue about this?
16.- In general the blots should be checked again “The photographs do not correspond to the same blots, therefore I recommend to present the correct ones. The bands from vinculin are oppose to LDHA in the last lane therefore I can assume that photographs are not from the same blot.”
17.- In the electropherogram I observed a difference in pick intensity, could this be an error of the reaction?
Additionally, the image of Perit Met and OPSmCC are identical.
Why HGSC was not included?
Author Response
1.-The authors mention “The right ovary and perimetrium presented hypoechoic nodules. Pelvic free fluid was present and Douglas peritoneum was thickened. Trans-abdominal ultrasonography evidenced a 4 cm para-aortic metastatic lymph node. Computed tomography (CT) scan confirmed the sonographic findings. The patient, due to the poor general condition, was unfit for surgery, therefore a transvaginal ultrasound-guided core needle biopsy of the left adnexal mass was performed. The pathological examination showed a high-grade carcinoma compatible with a gynecological origin.” However, the evidence was not shown, therefore, I recommend to add CT before and after to compare.
We are aware a prior CT scan would have been best to show, and therefore we fully understand the reviewer’s concern. Unfortunately, the only CT scan available was the one we show in the paper figure 1C, which is what we managed to retrieve in the search for both scans when we first drafted the manuscript. We decided to add the only scan available anyhow, but should the editor deem this not to be proper, we are willing to exclude these data from the figure and manuscript.
2.-The author mention “Mitotic figures were numerous, some with atypical features.” Can you show this features in the figure with an arrow.
This has been implemented, thank you.
3.-The sentence is redacted as if it had been done in the present work “Immunohistochemical staining showed negativity for WT-1, PAX-8 and estrogen receptor/progesterone receptor, excluding a tubo-ovarian high-grade serous carcinoma (HGSC) [3,8]”This was not evaluated in the author’s work
We have reported the immunohistochemical profile performed to exemplify the diagnostic algorithm useful for tumor characterization and differential diagnosis. To this aim, representative and essential immunohistochemistry figures were included rather than all those needed to perform diagnosis, which were however evaluated with the scope to reach a diagnostic consensus. To avoid confusion, for which we apologize, and being the diagnostic procedure beyond the scope of this work, we now rephrased the first paragraph of Results highlighting the findings that allowed the specific diagnosis, and currently shown in the figure.
4.- The authors mention “Synaptophysin and chromogranin were diffusely expressed (Fig. 1E), suggesting a neuroendocrine differentiation [9].” However, Synaptophysin was not evaluated in this work.
See the previous response.
5.- The authors mention “The tumor cells were also positive for pan-cytokeratin AE1/AE3, and TTF-1 (Fig. 1F), p53 (Fig. 1G).” But pan-cytokeratin AE1/AE3 was not evaluated in this work.
See the previous response.
6.- The authors mention “These morphological and immunohistochemical findings were similar to those of SCLC, but a metastatic origin was excluded by the CT scan.”
Can you explain how a CT determine metastatic origin of a tissue? It is the first time that I read this
asseveration.
In fact, the CT did not determine any metastatic origin, but it excluded this since it did not reveal a tumor in the lungs.
7.- The authors mention “The analysis allowed to detect the TP53 p.Y163C (c.488A>G, exon 5) missense mutation in all cancer specimens, correlated to a strong and diffuse p53 overexpression (Fig. 1G), confirming the already reported frequent alteration of TP53 in OPSmCCs.” Can you explain how a missense mutation correlates with protein expression? Does the p53 antibody recognize the amino acid change? Did you have antibodies that recognize p53 with the amino acid C and Y? The confirmation of TP53 alteration by the mutation could be suggested but it was not demonstrated in the present work.
It is common routinary practice in hospital pathology diagnostics to stain tumors with an anti-p53 antibody to ascertain overexpression of the protein, which is a readout for a mutation in the tumor suppressor gene. No antibodies are available or used against selected amino acid changes, unless they are post-translationally modified. Indeed, it is widely accepted and very well known that nearly all missense pathogenic mutations in p53 lead to a neomorphic pro-oncogenic function that increases stability and therefore is highlighted as an overexpression in the cancer tissues harboring the mutation. Parameters such as detection of a known pathogenic (hotspot, database-reported) p53 mutations, along with diffuse overexpression of the protein, as we performed, are considered sufficient to deem that the TSG is altered in cases such as ours.
8.- The authors mention “We performed a clustering analysis, using Pearson correlation as similarity measure, unveiling that “neuroendocrine” tumors have the most similar expression profile (Fig. 2A).” Can you describe and/or mention which tumors are those?
All the tumor classes are now available in Figure 2 legend. We applied a predictive test for tumor primary identification that was previously established in Ferracin’s lab (PMID: 34075699 and PMID: 21630269) to this rare SCOOPT sample. Here we present the outcome of this molecular test. We provided further clarifications in the manuscript.
9.- The authors mention “Among the miRNAs most expressed in OPSmCC, we detected miR-375, miR-141-3p, miR-200c-3p, miR-16-5p and miR-103a-3p (Fig. 2B). Did you compare the expression with a normal tissue to corroborate their increased expression?
The miRNAs we mentioned are the most expressed (increased would mean with respect to a control) in the analyzed OPSmCC (now SCOOPT) sample (Table 1). The expression level was evaluated by counting the copies of each miRNA by droplet digital PCR. The normalized expression of this sample, similarly to the others from the previous study (Laprovitera et al. PMID 34075699), was obtained by using SNORD44 as reference small RNA. Comparison, therefore, is not performed against a normal tissue, but the levels of the most expressed miRNAs are compared to the tumor samples described in Laprovitera et al. We provided a supplementary Table 1 with the miRNA expression in SCOOPT and 17 primary tumor classes, which was previously missing, and implemented the Materials and Methods section to improve clarity.
10.- The authors mention “MiR-141-3p and miR-200 family are up- 141 regulated in prostate cancers and regulate the metastatic process in many cancer types. In ovarian cancer the overexpression of miR-200 family has been associated with a mesothelial-to-epithelial transition and a more aggressive phenotype [14]. This evidence confirmed the similarity of OPSmCC to lung neuroendocrine carcinomas, despite the different site of origin.” Is the expression of miR-141-3p, miR-200 family are only up-regulated in neuroendrocrine carcinomas to confirm the similarity of OPSmCC to lung neuroendocrine carcinomas?
There was an error in this sentence, our apologies. We thank the reviewer for allowing us to spot this. The microRNA that points toward neuroendocrine carcinoma is miR-375. We fixed the sentence and provided miRNA expression data (Supplementary Table 1) to support the similarity between SCOOPT and both neuroendocrine and lung cancers, showing the whole profile is significantly more similar, rather than single miRNA families.
11.- The authors mention “PKM1 expression in OPSmCC and in the metastasis confirmed the similarity of this tumor with SCLC, suggesting that this tumor case may use predominantly glycolytic metabolic routes to sustain proliferation and growth.” The samples used are from ovary, therefore, you cannot confirm that they are similar to SCLC. Additionally, samples from peri-metastatic, OPSmCC and HGSC are similar and you pull apart or discuss only peri-metastatic and OPSmCC, therefore describing an unexisting mechanism regarding PKM1 protein.
We apologize to the reviewer for the lack of clarity in the description of this result. We agree that the samples used are from ovary but we would like to underline that, as we reported in Introduction section, the case that we analyzed (SCCOPT) is described in the literature as a very rare neuroendocrine carcinoma morphologically identical to small-cell lung cancer (SCLC). Further, since our results were obtained by dissecting one SCCOPT sample only we cannot generalize the findings for all the cases. Hence, to avoid overstatements we have modified the related text as follows: “PKM1 expression in SCCOPT and in PM suggests that at least this tumor case may use predominantly glycolytic metabolic routes to sustain proliferation and growth and hence may be similar to SCLC in terms of energetic profile”. To better clarify our results, we have added the densitometry values expressed as PKM1/PKM2 ratio in the Figure 3B.
12.- The authors mention “While glycolytic proteins were found up-regulated in the peritoneal metastasis (Fig. 3C), mitochondria oxidative proteins showed a marked reduction in both cancer tissues, suggesting the latter to have a low-OXPHOS signature and to rely on glycolysis [24] (Fig. 3D).” Except for CII-SDHB that is overexpressed in cancer tissues. You can argue and discuss why.
We thank the reviewer for the insightful comment. Indeed, the densitometric quantification supports the low-OXPHOS profile in SCCOPT and in PM samples for subunits belonging to OXPHOS complexes with both mtDNA and nDNA encoded subunits, while SDHB levels are increased. We can argue that this phenotype can be related either to exclusive nuclear control of CII subunits or to a prominent involvement of this enzyme in the TCA cycle that can be envisioned as fueled by glycolytic products.
13.- The authors mention “This was confirmed also by the high expression level of glucose transporter GLUT-1, as displayed by immunohistochemistry (Fig. 3E).”
You cannot confirm a low-OXPHOS signature in cancer tissue based in GLUT-1 expression showing only a photograph of OPSmCC.
We thank the reviewer, in agreement with whom we have added panel 3 (Fig. 3E) in which GLUT-1 expression in the non-cancer ovarian tissue of the SCCOPT patient is shown, to demonstrate that the transporter is indeed overexpressed in cancer cells. Although it is true that GLUT-1 expression is not sufficient per se to infer a low-OXPHOS signature, the latter is supported by several findings, whereby we have rephrased as follows: This was further corroborated by the high expression level of glucose transporter GLUT-1 compared to the adjacent non-cancer tissue, as displayed by immunohistochemistry analysis (Fig.3E).
14.- The authors mention “Intriguingly, respiratory complexes I (CI) and III (CIII) activity was higher in OPSmCC and in the metastasis compared to non-tumor tissue, revealing that oxidative metabolic routes may be functional in the analyzed case (Fig. 3F).” Can you explain why in non-tumor tissue the CI does not have activity? In this sentence the activity between OPSmCC, peri-metas and HGSC are similar, you are making a bias in the description of your results.
We thank the reviewer for this comment. We cannot exclude the lack of CI activity in NC-OV sample may be due to a very low activity of the enzyme or to a technical issue. Unfortunately, we cannot repeat these measurements because of the lack of the same (or other) non-cancer ovarian epithelium samples. Hence, to avoid any overstatements we decided to remove the activity data, and the corresponding text in Results, and Materials and Methods sections, from the revised version of the manuscript.
15.- The authors mention “Furthermore, the mutation was already reported in a patient with Charcot- Marie-Tooth but his potential pathogenetic nature remains uncertain [29]. Based on MT-ATP6 immunohistochemistry (Fig. 3H), the missense mutation did not seem to hamper the protein amount and the assembly of CV.”
How is the expression in the other tissues? You didn ́t include other tissues in your study. Can you argue about this?
We apologize to the reviewer for the lack of clarity here. In order not to incur in overstatements, since the expression level was not compared to other tissues, and the aim here was to verify whether the mutation may affect protein synthesis and CV assembly, we rephrased as follows: Based on a positive MT-ATP6 immunohistochemical staining (Fig. 3G), the missense mutation did not seem to cause an impairment in the subunit synthesis, and complex V (CV) resulted fully assembled (Fig. 3H).
16.- In general the blots should be checked again “The photographs do not correspond to the same blots, therefore I recommend to present the correct ones. The bands from vinculin are oppose to LDHA in the last lane therefore I can assume that photographs are not from the same blot.”
We apologize to the reviewer for overlooking this, as we reported the vinculin taken from a second blot with the same loading. The vinculin from the same blot of LDHA has now been reported in Fig. 3C.
17.- In the electropherogram I observed a difference in pick intensity, could this be an error of the reaction? Additionally, the image of Perit Met and OPSmCC are identical. Why HGSC was not included?
We are not sure we understand here what the reviewer is referring to. Whether to the picture intensity in terms of color, or the peaks height (which does not reflect intensity). If the reviewer is referring to the different height of the peaks, this is random within every electropherogram in Sanger sequencing (and due, for instance, to presence of residues of ethanol after washing plates) but does not affect the base call. Errors of reaction, moreover, are ruled out on a different DNA extraction, PCR and sequence, which in this case were performed on the peritoneal metastasis, as well as on the primary tumor, revealing the same mutation, which did not occur in the non-cancer matched tissue. This indicated the mutation to be cancer-specific, confirming the peritoneal metastasis originated by a mtDNA mutated clone of the primary, with no need for a non-matched tumor control (HGSC).
Reviewer 3 Report
MicroRNA and metabolic profiling of a primary ovarian neuro- 2 endocrine carcinoma pulmonary-type reveals a high degree of 3 similarity with small cell lung cancer
I have carefully reviewed the article and found that the paper has merit for publication; however, there are few important points that need to be addressed or justified by the authors.
-
The objectives of the study stated in the introduction does not match with the findings and title of the study where the title emphasizes on mirna profiling and metabolic profiling but in the introduction author states focus on mirna profiling and molecular biology/genetics techniques.
-
The introduction written can be improved.
-
This study focused on one patient diagnosed with OPSmCC. From my understanding, authors used the OPSmCC samples together with another 79 samples from 17 different cancer sites for mirna profiling (please refer to figure 2(a)).. Major concern is that will the control tissue used as calibrator in the q-PCR analysis be suitable to be compared with different types of cancer tissue? Will the idea of using “pool” normal samples be a better idea for comparison? How many control/normal samples the authors have? Did samples other than OPSmCC receive approval to conduct study from the ethics committee?
-
Mir-375, Mir-200 and mir-141-5p are the 3 mirnas that are highly overexpressed in the analysis. However, the author did not clearly state the actual relationship of these 3 mirnas with the downstream pathway mentioned which is the PI3K/Akt/mTOR pathway. On what basis identification of these mirna and their targets were performed? Did the author use any specific database? The findings and discussion between identified mirnas with PI3K/Akt/mTOR pathway-glycolytic metabolism gene targets seems disconnected.
-
The conclusion written does not reflect the actual study performed.
-
In the materials and methods section Mirna profiling analysis author stated; Prediction of the tissue-of-origin was performed as previously described. Can the author justify why this has to be performed using an algorithm?
It is hoped that authors will be able to justify these queries. Thank you.
Author Response
1.The objectives of the study stated in the introduction does not match with the findings and title of the study where the title emphasizes on mirna profiling and metabolic profiling but in the introduction author states focus on mirna profiling and molecular biology/genetics techniques.
To improve clarity, we modified the introduction stating that we indeed exploit molecular biology/genetics techniques to infer the metabolic features of the tumor.
2.The introduction written can be improved.
We apologize to the reviewer if he/she did not find the introduction suitable. Unfortunately, without a detailed indication of what could be improved it is difficult to act on the text, also in light of the positive comments of reviewer 1. We do hope that the modifications introduced to better match the objective stated, techniques applied and the title/results, few as they are, may be the cogent point that needed to be addressed to render the introduction acceptable to this reviewer.
3.This study focused on one patient diagnosed with OPSmCC. From my understanding, authors used the OPSmCC samples together with another 79 samples from 17 different cancer sites for mirna profiling (please refer to figure 2(a)). Major concern is that will the control tissue used as calibrator in the q-PCR analysis be suitable to be compared with different types of cancer tissue? Will the idea of using “pool” normal samples be a better idea for comparison? How many control/normal samples the authors have? Did samples other than OPSmCC receive approval to conduct study from the ethics committee?
We are now aware that some sentences were not clear in the original case-report version of the manuscript, for which we apologize. We provided further clarifications in the Materials and Methods section about how the miRNA test was performed. We applied a predictive test for tumor primary identification that was previously established in Ferracin’s lab (PMID: 34075699 and PMID: 21630269) to this rare OPSmCC (now indicated as SCOOPT, as suggested by reviewer 1) sample. Here we present the outcome of this molecular test. The miRNAs we mentioned are the most expressed in the analyzed SCOOPT sample. The expression level was evaluated by counting the copies of each miRNA by droplet digital PCR. The normalized expression of this sample, similarly to the others from the previous study (Laprovitera et al. PMID 34075699), was obtained by using SNORD44 as reference small RNA. We provided a supplemental Table with the miRNA expression in SCOOPT and 17 primary tumor classes (average value of 4-6 samples). We would like to remark that for our prediction analysis we did not use any normal tissue or control samples as intended by the reviewer. With respect to the tumor tissues used to train the algorithm in Laprovitera et al, they were collected in accordance with the Declaration of Helsinki, and following a protocol approved by the Ethics Committee Center Emilia-Romagna Region—Italy (protocol 130/2016/U/Tess).
4.Mir-375, Mir-200 and mir-141-5p are the 3 mirnas that are highly overexpressed in the analysis. However, the author did not clearly state the actual relationship of these 3 mirnas with the downstream pathway mentioned which is the PI3K/Akt/mTOR pathway. On what basis identification of these mirna and their targets were performed? Did the author use any specific database? The findings and discussion between identified mirnas with PI3K/Akt/mTOR pathway-glycolytic metabolism gene targets seems disconnected.
The microRNA profile was obtained to perform the similarity assessment and predictive analysis. We did not perform a miRNA-target analysis. As per the text introducing paragraph 2 of the results, now slightly modified, and on the basis of the outcome of the predictive analysis, we decided to investigate the PI3K/Akt/mTOR pathway, since this is a feature of SCLC that may provide indications for therapies such as those based on mTOR inhibitors (Everolimus, for instance, which we comment on in the Manuscript). Therefore, the reviewer is correct that there is no direct link to be found in the paper between specific miRNA and the mTOR pathway, which we investigated to verify if other features than miRNA profiling pointed in the same direction as SCLC.
5.The conclusion written does not reflect the actual study performed.
We have now revised the conclusion to make it more adherent with the study and the results reported, hoping the reviewer will find it more suitable than the previous version.
6.In the materials and methods section Mirna profiling analysis author stated; Prediction of the tissue-of-origin was performed as previously described. Can the author justify why this has to be performed using an algorithm?
We are not sure we understand this reviewer’s question. The reason for the use of an algorithm is already described in the original paper where it was generated (Laprovitera et al. Molecular Oncology 2021). Basically, the prediction is obtained by combining LASSO and the shrunken centroids approach. We did not report here the detailed predictive procedure since it is already published.
Round 2
Reviewer 1 Report
The authors have revised the manuscript significantly. However, please verify the GenBank accession, as they don't seem to be made public yet. GenBank Accession Numbers OP342772-OP342773.
Author Response
The authors have revised the manuscript significantly. However, please verify the GenBank accession, as they don't seem to be made public yet. GenBank Accession Numbers OP342772-OP342773.
Response: Yes, unfortunately this depends on the queue GenBank has. We apologize for not being able to help. We will communicate to the Journal as soon as GenBank lets us know that the accessions have been made public.
Reviewer 2 Report
1.-The authors mention “The right ovary and perimetrium presented hypoechoic nodules. Pelvic free fluid was present and Douglas peritoneum was thickened. Trans-abdominal ultrasonography evidenced a 4 cm para-aortic metastatic lymph node. Computed tomography (CT) scan confirmed the sonographic findings. The patient, due to the poor general condition, was unfit for surgery, therefore a transvaginal ultrasound-guided core needle biopsy of the left adnexal mass was performed. The pathological examination showed a high-grade carcinoma compatible with a gynecological origin.” However, the evidence was not shown, therefore, I recommend to add CT before and after to compare.
We are aware a prior CT scan would have been best to show, and therefore we fully understand the reviewer’s concern. Unfortunately, the only CT scan available was the one we show in the paper figure 1C, which is what we managed to retrieve in the search for both scans when we first drafted the manuscript. We decided to add the only scan available anyhow, but should the editor deem this not to be proper, we are willing to exclude these data from the figure and manuscript.
Response: The text can be extended and mention a CT of normal tissue features to remark the carcinoma finding.
2.-The author mention “Mitotic figures were numerous, some with atypical features.” Can you show this features in the figure with an arrow.
This has been implemented, thank you.
Response: “The mitotic figures are not clear” please check Journal of Cardiothoracic Surgery 5(1):115.
3.-The sentence is redacted as if it had been done in the present work “Immunohistochemical staining showed negativity for WT-1, PAX-8 and estrogen receptor/progesterone receptor, excluding a tubo-ovarian high-grade serous carcinoma (HGSC) [3,8]”This was not evaluated in the author’s work
We have reported the immunohistochemical profile performed to exemplify the diagnostic algorithm useful for tumor characterization and differential diagnosis. To this aim, representative and essential immunohistochemistry figures were included rather than all those needed to perform diagnosis, which were however evaluated with the scope to reach a diagnostic consensus. To avoid confusion, for which we apologize, and being the diagnostic procedure beyond the scope of this work, we now rephrased the first paragraph of Results highlighting the findings that allowed the specific diagnosis, and currently shown in the figure.
Response: As the paper do not have limit to show results, I recommend to exclude from the manuscript the experiments not showed.
4.- The authors mention “Synaptophysin and chromogranin were diffusely expressed (Fig. 1E), suggesting a neuroendocrine differentiation [9].” However, Synaptophysin was not evaluated in this work.
See the previous response.
Response: See the previous response.
5.- The authors mention “The tumor cells were also positive for pan-cytokeratin AE1/AE3, and TTF-1 (Fig. 1F), p53 (Fig. 1G).” But pan-cytokeratin AE1/AE3 was not evaluated in this work.
See the previous response.
Response: See the previous response.
6.- The authors mention “These morphological and immunohistochemical findings were similar to those of SCLC, but a metastatic origin was excluded by the CT scan.”
Can you explain how a CT determine metastatic origin of a tissue? It is the first time that I read this
asseveration.
In fact, the CT did not determine any metastatic origin, but it excluded this since it did not reveal a tumor in the lungs.
Response: If CT did not determine any metastatic origin can you omitted from the sentence.
7.- The authors mention “The analysis allowed to detect the TP53 p.Y163C (c.488A>G, exon 5) missense mutation in all cancer specimens, correlated to a strong and diffuse p53 overexpression (Fig. 1G), confirming the already reported frequent alteration of TP53 in OPSmCCs.” Can you explain how a missense mutation correlates with protein expression? Does the p53 antibody recognize the amino acid change? Did you have antibodies that recognize p53 with the amino acid C and Y? The confirmation of TP53 alteration by the mutation could be suggested but it was not demonstrated in the present work.
It is common routinary practice in hospital pathology diagnostics to stain tumors with an anti-p53 antibody to ascertain overexpression of the protein, which is a readout for a mutation in the tumor suppressor gene. No antibodies are available or used against selected amino acid changes, unless they are post-translationally modified. Indeed, it is widely accepted and very well known that nearly all missense pathogenic mutations in p53 lead to a neomorphic pro-oncogenic function that increases stability and therefore is highlighted as an overexpression in the cancer tissues harboring the mutation. Parameters such as detection of a known pathogenic (hotspot, database-reported) p53 mutations, along with diffuse overexpression of the protein, as we performed, are considered sufficient to deem that the TSG is altered in cases such as ours.
Response: If a missense mutation occur it supposed to reflect an amino acid change in protein sequence causing a post-translation modification in reference to your comment “No antibodies are available or used against selected amino acid changes, unless they are post-translationally modified.”
I agree with this comment “Indeed, it is widely accepted and very well known that nearly all missense pathogenic mutations in p53 lead to a neomorphic pro-oncogenic function that increases stability and therefore is highlighted as an overexpression in the cancer tissues harboring the mutation.” Only if you compare the expression of p53 in normal tissue versus tumor tissue. It can not assume that all the normal tissues express null p53 and tumor expresses p53 it has to be demonstrated. Therefore, I suggest to include p53 expression in NC-OV, Perit Met, and HGSC or least NC-OV and OPSmCCs to conclude that p53 is overexpressed in OPSmCCs.
Regarding the comment “Parameters such as detection of a known pathogenic (hotspot, database-reported) p53 mutations, along with diffuse overexpression of the protein, as we performed, are considered sufficient to deem that the TSG is altered in cases such as ours.” If you sequenced p53 and the only mutation, did you find was “TP53 p.Y163C (c.488A>G, exon 5)” hence you can suggest that this mutation is responsible of p53 overexpression, however, you are missing a clear demonstration of p53 overexpression.
8.- The authors mention “We performed a clustering analysis, using Pearson correlation as similarity measure, unveiling that “neuroendocrine” tumors have the most similar expression profile (Fig. 2A).” Can you describe and/or mention which tumors are those?
All the tumor classes are now available in Figure 2 legend. We applied a predictive test for tumor primary identification that was previously established in Ferracin’s lab (PMID: 34075699 and PMID: 21630269) to this rare SCOOPT sample. Here we present the outcome of this molecular test. We provided further clarifications in the manuscript.
Response:
9.- The authors mention “Among the miRNAs most expressed in OPSmCC, we detected miR-375, miR-141-3p, miR-200c-3p, miR-16-5p and miR-103a-3p (Fig. 2B). Did you compare the expression with a normal tissue to corroborate their increased expression?
The miRNAs we mentioned are the most expressed (increased would mean with respect to a control) in the analyzed OPSmCC (now SCOOPT) sample (Table 1). The expression level was evaluated by counting the copies of each miRNA by droplet digital PCR. The normalized expression of this sample, similarly to the others from the previous study (Laprovitera et al. PMID 34075699), was obtained by using SNORD44 as reference small RNA. Comparison, therefore, is not performed against a normal tissue, but the levels of the most expressed miRNAs are compared to the tumor samples described in Laprovitera et al. We provided a supplementary Table 1 with the miRNA expression in SCOOPT and 17 primary tumor classes, which was previously missing, and implemented the Materials and Methods section to improve clarity.
Response: The expression of these miRNAs can be discussed with expression in others tissues and as well as with normal tissue like in Laprovitera et al. 2021 study.
10.- The authors mention “MiR-141-3p and miR-200 family are up- 141 regulated in prostate cancers and regulate the metastatic process in many cancer types. In ovarian cancer the overexpression of miR-200 family has been associated with a mesothelial-to-epithelial transition and a more aggressive phenotype [14]. This evidence confirmed the similarity of OPSmCC to lung neuroendocrine carcinomas, despite the different site of origin.” Is the expression of miR-141-3p, miR-200 family are only up-regulated in neuroendrocrine carcinomas to confirm the similarity of OPSmCC to lung neuroendocrine carcinomas?
There was an error in this sentence, our apologies. We thank the reviewer for allowing us to spot this. The microRNA that points toward neuroendocrine carcinoma is miR-375. We fixed the sentence and provided miRNA expression data (Supplementary Table 1) to support the similarity between SCOOPT and both neuroendocrine and lung cancers, showing the whole profile is significantly more similar, rather than single miRNA families.
Response: The correction is asserted.
11.- The authors mention “PKM1 expression in OPSmCC and in the metastasis confirmed the similarity of this tumor with SCLC, suggesting that this tumor case may use predominantly glycolytic metabolic routes to sustain proliferation and growth.” The samples used are from ovary, therefore, you cannot confirm that they are similar to SCLC. Additionally, samples from peri-metastatic, OPSmCC and HGSC are similar and you pull apart or discuss only peri-metastatic and OPSmCC, therefore describing an unexisting mechanism regarding PKM1 protein.
We apologize to the reviewer for the lack of clarity in the description of this result. We agree that the samples used are from ovary but we would like to underline that, as we reported in Introduction section, the case that we analyzed (SCCOPT) is described in the literature as a very rare neuroendocrine carcinoma morphologically identical to small-cell lung cancer (SCLC). Further, since our results were obtained by dissecting one SCCOPT sample only we cannot generalize the findings for all the cases. Hence, to avoid overstatements we have modified the related text as follows: “PKM1 expression in SCCOPT and in PM suggests that at least this tumor case may use predominantly glycolytic metabolic routes to sustain proliferation and growth and hence may be similar to SCLC in terms of energetic profile”. To better clarify our results, we have added the densitometry values expressed as PKM1/PKM2 ratio in the Figure 3B.
Response: The correction is asserted.
12.- The authors mention “While glycolytic proteins were found up-regulated in the peritoneal metastasis (Fig. 3C), mitochondria oxidative proteins showed a marked reduction in both cancer tissues, suggesting the latter to have a low-OXPHOS signature and to rely on glycolysis [24] (Fig. 3D).” Except for CII-SDHB that is overexpressed in cancer tissues. You can argue and discuss why.
We thank the reviewer for the insightful comment. Indeed, the densitometric quantification supports the low-OXPHOS profile in SCCOPT and in PM samples for subunits belonging to OXPHOS complexes with both mtDNA and nDNA encoded subunits, while SDHB levels are increased. We can argue that this phenotype can be related either to exclusive nuclear control of CII subunits or to a prominent involvement of this enzyme in the TCA cycle that can be envisioned as fueled by glycolytic products.
Response: The correction is asserted.
13.- The authors mention “This was confirmed also by the high expression level of glucose transporter GLUT-1, as displayed by immunohistochemistry (Fig. 3E).”
You cannot confirm a low-OXPHOS signature in cancer tissue based in GLUT-1 expression showing only a photograph of OPSmCC.
We thank the reviewer, in agreement with whom we have added panel 3 (Fig. 3E) in which GLUT-1 expression in the non-cancer ovarian tissue of the SCCOPT patient is shown, to demonstrate that the transporter is indeed overexpressed in cancer cells. Although it is true that GLUT-1 expression is not sufficient per se to infer a low-OXPHOS signature, the latter is supported by several findings, whereby we have rephrased as follows: This was further corroborated by the high expression level of glucose transporter GLUT-1 compared to the adjacent non-cancer tissue, as displayed by immunohistochemistry analysis (Fig.3E).
Response: The correction is asserted.
14.- The authors mention “Intriguingly, respiratory complexes I (CI) and III (CIII) activity was higher in OPSmCC and in the metastasis compared to non-tumor tissue, revealing that oxidative metabolic routes may be functional in the analyzed case (Fig. 3F).” Can you explain why in non-tumor tissue the CI does not have activity? In this sentence the activity between OPSmCC, peri-metas and HGSC are similar, you are making a bias in the description of your results.
We thank the reviewer for this comment. We cannot exclude the lack of CI activity in NC-OV sample may be due to a very low activity of the enzyme or to a technical issue. Unfortunately, we cannot repeat these measurements because of the lack of the same (or other) non-cancer ovarian epithelium samples. Hence, to avoid any overstatements we decided to remove the activity data, and the corresponding text in Results, and Materials and Methods sections, from the revised version of the manuscript.
15.- The authors mention “Furthermore, the mutation was already reported in a patient with Charcot- Marie-Tooth but his potential pathogenetic nature remains uncertain [29]. Based on MT-ATP6 immunohistochemistry (Fig. 3H), the missense mutation did not seem to hamper the protein amount and the assembly of CV.”
How is the expression in the other tissues? You didn ́t include other tissues in your study. Can you argue about this?
We apologize to the reviewer for the lack of clarity here. In order not to incur in overstatements, since the expression level was not compared to other tissues, and the aim here was to verify whether the mutation may affect protein synthesis and CV assembly, we rephrased as follows: Based on a positive MT-ATP6 immunohistochemical staining (Fig. 3G), the missense mutation did not seem to cause an impairment in the subunit synthesis, and complex V (CV) resulted fully assembled (Fig. 3H).
Response: A non-cancer ovarian tissue should be added to compare.
16.- In general the blots should be checked again “The photographs do not correspond to the same blots, therefore I recommend to present the correct ones. The bands from vinculin are oppose to LDHA in the last lane therefore I can assume that photographs are not from the same blot.”
We apologize to the reviewer for overlooking this, as we reported the vinculin taken from a second blot with the same loading. The vinculin from the same blot of LDHA has now been reported in Fig. 3C.
Response: Blot A as it can be seen the bands runs similar and vinculin and S6K can said that are from the same membrane. However, in Blot B Vinculin looks right while in PKM2 an inclination can be appreciate since SCCOPT continuing and being more pronounce in HGSC line. In the same concern it can be seen that vinculin of the upper panel has a up inclination in the right corner while the band of PKM1 do not has it. I recommend to check in deep your blots to assure that are correct. Actually, I recommend to send the complete Blot with out cutout.
17.- In the electropherogram I observed a difference in pick intensity, could this be an error of the reaction? Additionally, the image of Perit Met and OPSmCC are identical. Why HGSC was not included?
We are not sure we understand here what the reviewer is referring to. Whether to the picture intensity in terms of color, or the peaks height (which does not reflect intensity). If the reviewer is referring to the different height of the peaks, this is random within every electropherogram in Sanger sequencing (and due, for instance, to presence of residues of ethanol after washing plates) but does not affect the base call. Errors of reaction, moreover, are ruled out on a different DNA extraction, PCR and sequence, which in this case were performed on the peritoneal metastasis, as well as on the primary tumor, revealing the same mutation, which did not occur in the non-cancer matched tissue. This indicated the mutation to be cancer-specific, confirming the peritoneal metastasis originated by a mtDNA mutated clone of the primary, with no need for a non-matched tumor control (HGSC).
Response: If you don’t analyze HGSC you cannot conclude that is a cancer-specific mutation, because it can be a general cancer-mutation, therefore, I recommend to add the HGSC sequencing sample .
Author Response
1.-The authors mention “The right ovary and perimetrium presented hypoechoic nodules. Pelvic free fluid was present and Douglas peritoneum was thickened. Trans-abdominal ultrasonography evidenced a 4 cm para-aortic metastatic lymph node. Computed tomography (CT) scan confirmed the sonographic findings. The patient, due to the poor general condition, was unfit for surgery, therefore a transvaginal ultrasound-guided core needle biopsy of the left adnexal mass was performed. The pathological examination showed a high-grade carcinoma compatible with a gynecological origin.” However, the evidence was not shown, therefore, I recommend to add CT before and after to compare.
We are aware a prior CT scan would have been best to show, and therefore we fully understand the reviewer’s concern. Unfortunately, the only CT scan available was the one we show in the paper figure 1C, which is what we managed to retrieve in the search for both scans when we first drafted the manuscript. We decided to add the only scan available anyhow, but should the editor deem this not to be proper, we are willing to exclude these data from the figure and manuscript.
Response: The text can be extended and mention a CT of normal tissue features to remark the carcinoma finding.
Response: CTs of healthy individuals/organs are negative, i.e. they do not show lesions. As it is evident that a normal CT from any healthy subject is not a feasible add-on to the paper, we modified the text to make it clearer, as follows: “but a metastatic origin was excluded by the lack of lung lesions in the CT scan.”
2.-The author mention “Mitotic figures were numerous, some with atypical features.” Can you show this features in the figure with an arrow.
This has been implemented, thank you.
Response: “The mitotic figures are not clear” please check Journal of Cardiothoracic Surgery 5(1):115.
Response: We apologize with the reviewer. The figure has been improved accordingly.
3.-The sentence is redacted as if it had been done in the present work “Immunohistochemical staining showed negativity for WT-1, PAX-8 and estrogen receptor/progesterone receptor, excluding a tubo-ovarian high-grade serous carcinoma (HGSC) [3,8]”This was not evaluated in the author’s work
We have reported the immunohistochemical profile performed to exemplify the diagnostic algorithm useful for tumor characterization and differential diagnosis. To this aim, representative and essential immunohistochemistry figures were included rather than all those needed to perform diagnosis, which were however evaluated with the scope to reach a diagnostic consensus. To avoid confusion, for which we apologize, and being the diagnostic procedure beyond the scope of this work, we now rephrased the first paragraph of Results highlighting the findings that allowed the specific diagnosis, and currently shown in the figure.
Response: As the paper do not have limit to show results, I recommend to exclude from the manuscript the experiments not showed.
Response: We do not agree with the reviewer on this. We fail to see what could be the issue in stating that routinary diagnostic stainings are data not shown, as this is not a pathology-based paper. We have reported the data needed to define the SCOOPT diagnosis, while the remaining (not shown) tell the story of how we achieved the final diagnosis.
4.- The authors mention “Synaptophysin and chromogranin were diffusely expressed (Fig. 1E), suggesting a neuroendocrine differentiation [9].” However, Synaptophysin was not evaluated in this work.
See the previous response.
Response: See the previous response.
Response: see previously.
5.- The authors mention “The tumor cells were also positive for pan-cytokeratin AE1/AE3, and TTF-1 (Fig. 1F), p53 (Fig. 1G).” But pan-cytokeratin AE1/AE3 was not evaluated in this work.
See the previous response.
Response: See the previous response.
Response: see previously.
6.- The authors mention “These morphological and immunohistochemical findings were similar to those of SCLC, but a metastatic origin was excluded by the CT scan.”
Can you explain how a CT determine metastatic origin of a tissue? It is the first time that I read this
asseveration.
In fact, the CT did not determine any metastatic origin, but it excluded this since it did not reveal a tumor in the lungs.
Response: If CT did not determine any metastatic origin can you omitted from the sentence.
Response: We did not state, as we attempted to explain previously, that CT did not determine any metastatic origin in the sense that it was useless in helping to determine if the tumor was metastatic or not. The sentence has a different meaning: the CT was useful in ruling out the metastatic origin, and this is relevant information we do not feel can be omitted from the paper. Perhaps the current rephrasing of the sentence (see Response to 1) better helps clearing this.
7.- The authors mention “The analysis allowed to detect the TP53 p.Y163C (c.488A>G, exon 5) missense mutation in all cancer specimens, correlated to a strong and diffuse p53 overexpression (Fig. 1G), confirming the already reported frequent alteration of TP53 in OPSmCCs.” Can you explain how a missense mutation correlates with protein expression? Does the p53 antibody recognize the amino acid change? Did you have antibodies that recognize p53 with the amino acid C and Y? The confirmation of TP53 alteration by the mutation could be suggested but it was not demonstrated in the present work.
It is common routinary practice in hospital pathology diagnostics to stain tumors with an anti-p53 antibody to ascertain overexpression of the protein, which is a readout for a mutation in the tumor suppressor gene. No antibodies are available or used against selected amino acid changes, unless they are post-translationally modified. Indeed, it is widely accepted and very well known that nearly all missense pathogenic mutations in p53 lead to a neomorphic pro-oncogenic function that increases stability and therefore is highlighted as an overexpression in the cancer tissues harboring the mutation. Parameters such as detection of a known pathogenic (hotspot, database-reported) p53 mutations, along with diffuse overexpression of the protein, as we performed, are considered sufficient to deem that the TSG is altered in cases such as ours.
Response: If a missense mutation occur it supposed to reflect an amino acid change in protein sequence causing a post-translation modification in reference to your comment “No antibodies are available or used against selected amino acid changes, unless they are post-translationally modified.”
Response: No, all missense mutations cause a change in the protein amino acid (during protein translation), but only few such amino acids are sites for post-translational modifications such as phosphorylation, SUMOylation, succinylation, glutathionylation, and so forth. A missense mutation that changes an amino acid that is not, for instance, a Serine or a Threonine that are target for phosphorylation by specific kinases may seldom find an available antibody to bind. At the same time pathogenic changes are not only those of amino acids that become post-translationally modified.
I agree with this comment “Indeed, it is widely accepted and very well known that nearly all missense pathogenic mutations in p53 lead to a neomorphic pro-oncogenic function that increases stability and therefore is highlighted as an overexpression in the cancer tissues harboring the mutation.” Only if you compare the expression of p53 in normal tissue versus tumor tissue. It can not assume that all the normal tissues express null p53 and tumor expresses p53 it has to be demonstrated. Therefore, I suggest to include p53 expression in NC-OV, Perit Met, and HGSC or least NC-OV and OPSmCCs to conclude that p53 is overexpressed in OPSmCCs.
Response: Please see previous and following replies.
Regarding the comment “Parameters such as detection of a known pathogenic (hotspot, database-reported) p53 mutations, along with diffuse overexpression of the protein, as we performed, are considered sufficient to deem that the TSG is altered in cases such as ours.” If you sequenced p53 and the only mutation, did you find was “TP53 p.Y163C (c.488A>G, exon 5)” hence you can suggest that this mutation is responsible of p53 overexpression, however, you are missing a clear demonstration of p53 overexpression.
Response: We did sequence p53 as stated and reported the only mutation found. This mutation is a known pathogenic hotspot, as we reported: Such TP53 mutation is classified as “Pathogenic” according to ACMG (American College of Medical Genetics) classification and ClinVar archive (https://varsome.com/, last access: 27th July 2022). A marked positive staining of the protein, not alone but along with the established data on its pathogenicity, allows us to tell the tumor has a p53 pathogenic/driver mutation, which is really all that is relevant to the story. We never stated or attempted to demonstrate that the mutation (already known as pathogenic) is responsibile for the protein overexpression. As a pathogenic hotspot, this correlation is not novel. The term ‘overexpressed’ does not require a control according to the pathology guidelines as outlined in Köbel M, Kang EY. The Many Uses of p53 Immunohistochemistry in Gynecological Pathology: Proceedings of the ISGyP Companion Society Session at the 2020 USCAP Annual9 Meeting. Int J Gynecol Pathol. 2021;40(1):32-40. doi:10.1097/PGP.0000000000000725. It remains valid and we would like to remark that the causal relation between the hotspot mutation and the degree of the protein expression is of no scope to this paper; it is the detection of the mutation, corroborated by IHC, that points to the diagnosis.
8.- The authors mention “We performed a clustering analysis, using Pearson correlation as similarity measure, unveiling that “neuroendocrine” tumors have the most similar expression profile (Fig. 2A).” Can you describe and/or mention which tumors are those?
All the tumor classes are now available in Figure 2 legend. We applied a predictive test for tumor primary identification that was previously established in Ferracin’s lab (PMID: 34075699 and PMID: 21630269) to this rare SCOOPT sample. Here we present the outcome of this molecular test. We provided further clarifications in the manuscript.
Response:
9.- The authors mention “Among the miRNAs most expressed in OPSmCC, we detected miR-375, miR-141-3p, miR-200c-3p, miR-16-5p and miR-103a-3p (Fig. 2B). Did you compare the expression with a normal tissue to corroborate their increased expression?
The miRNAs we mentioned are the most expressed (increased would mean with respect to a control) in the analyzed OPSmCC (now SCOOPT) sample (Table 1). The expression level was evaluated by counting the copies of each miRNA by droplet digital PCR. The normalized expression of this sample, similarly to the others from the previous study (Laprovitera et al. PMID 34075699), was obtained by using SNORD44 as reference small RNA. Comparison, therefore, is not performed against a normal tissue, but the levels of the most expressed miRNAs are compared to the tumor samples described in Laprovitera et al. We provided a supplementary Table 1 with the miRNA expression in SCOOPT and 17 primary tumor classes, which was previously missing, and implemented the Materials and Methods section to improve clarity.
Response: The expression of these miRNAs can be discussed with expression in others tissues and as well as with normal tissue like in Laprovitera et al. 2021 study.
Response: In this study we applied the same analysis as in Laprovitera et al, which allows us to remark that normal samples are not available for a comparison of expression, as previously stated (please see the response to this remark provided during the revision round 1). The comparison with other primary tumors is, on the other hand, reported in supplementary Table 1.
10.- The authors mention “MiR-141-3p and miR-200 family are up- 141 regulated in prostate cancers and regulate the metastatic process in many cancer types. In ovarian cancer the overexpression of miR-200 family has been associated with a mesothelial-to-epithelial transition and a more aggressive phenotype [14]. This evidence confirmed the similarity of OPSmCC to lung neuroendocrine carcinomas, despite the different site of origin.” Is the expression of miR-141-3p, miR-200 family are only up-regulated in neuroendrocrine carcinomas to confirm the similarity of OPSmCC to lung neuroendocrine carcinomas?
There was an error in this sentence, our apologies. We thank the reviewer for allowing us to spot this. The microRNA that points toward neuroendocrine carcinoma is miR-375. We fixed the sentence and provided miRNA expression data (Supplementary Table 1) to support the similarity between SCOOPT and both neuroendocrine and lung cancers, showing the whole profile is significantly more similar, rather than single miRNA families.
Response: The correction is asserted.
11.- The authors mention “PKM1 expression in OPSmCC and in the metastasis confirmed the similarity of this tumor with SCLC, suggesting that this tumor case may use predominantly glycolytic metabolic routes to sustain proliferation and growth.” The samples used are from ovary, therefore, you cannot confirm that they are similar to SCLC. Additionally, samples from peri-metastatic, OPSmCC and HGSC are similar and you pull apart or discuss only peri-metastatic and OPSmCC, therefore describing an unexisting mechanism regarding PKM1 protein.
We apologize to the reviewer for the lack of clarity in the description of this result. We agree that the samples used are from ovary but we would like to underline that, as we reported in Introduction section, the case that we analyzed (SCCOPT) is described in the literature as a very rare neuroendocrine carcinoma morphologically identical to small-cell lung cancer (SCLC). Further, since our results were obtained by dissecting one SCCOPT sample only we cannot generalize the findings for all the cases. Hence, to avoid overstatements we have modified the related text as follows: “PKM1 expression in SCCOPT and in PM suggests that at least this tumor case may use predominantly glycolytic metabolic routes to sustain proliferation and growth and hence may be similar to SCLC in terms of energetic profile”. To better clarify our results, we have added the densitometry values expressed as PKM1/PKM2 ratio in the Figure 3B.
Response: The correction is asserted.
12.- The authors mention “While glycolytic proteins were found up-regulated in the peritoneal metastasis (Fig. 3C), mitochondria oxidative proteins showed a marked reduction in both cancer tissues, suggesting the latter to have a low-OXPHOS signature and to rely on glycolysis [24] (Fig. 3D).” Except for CII-SDHB that is overexpressed in cancer tissues. You can argue and discuss why.
We thank the reviewer for the insightful comment. Indeed, the densitometric quantification supports the low-OXPHOS profile in SCCOPT and in PM samples for subunits belonging to OXPHOS complexes with both mtDNA and nDNA encoded subunits, while SDHB levels are increased. We can argue that this phenotype can be related either to exclusive nuclear control of CII subunits or to a prominent involvement of this enzyme in the TCA cycle that can be envisioned as fueled by glycolytic products.
Response: The correction is asserted.
13.- The authors mention “This was confirmed also by the high expression level of glucose transporter GLUT-1, as displayed by immunohistochemistry (Fig. 3E).”
You cannot confirm a low-OXPHOS signature in cancer tissue based in GLUT-1 expression showing only a photograph of OPSmCC.
We thank the reviewer, in agreement with whom we have added panel 3 (Fig. 3E) in which GLUT-1 expression in the non-cancer ovarian tissue of the SCCOPT patient is shown, to demonstrate that the transporter is indeed overexpressed in cancer cells. Although it is true that GLUT-1 expression is not sufficient per se to infer a low-OXPHOS signature, the latter is supported by several findings, whereby we have rephrased as follows: This was further corroborated by the high expression level of glucose transporter GLUT-1 compared to the adjacent non-cancer tissue, as displayed by immunohistochemistry analysis (Fig.3E).
Response: The correction is asserted.
14.- The authors mention “Intriguingly, respiratory complexes I (CI) and III (CIII) activity was higher in OPSmCC and in the metastasis compared to non-tumor tissue, revealing that oxidative metabolic routes may be functional in the analyzed case (Fig. 3F).” Can you explain why in non-tumor tissue the CI does not have activity? In this sentence the activity between OPSmCC, peri-metas and HGSC are similar, you are making a bias in the description of your results.
We thank the reviewer for this comment. We cannot exclude the lack of CI activity in NC-OV sample may be due to a very low activity of the enzyme or to a technical issue. Unfortunately, we cannot repeat these measurements because of the lack of the same (or other) non-cancer ovarian epithelium samples. Hence, to avoid any overstatements we decided to remove the activity data, and the corresponding text in Results, and Materials and Methods sections, from the revised version of the manuscript.
15.- The authors mention “Furthermore, the mutation was already reported in a patient with Charcot- Marie-Tooth but his potential pathogenetic nature remains uncertain [29]. Based on MT-ATP6 immunohistochemistry (Fig. 3H), the missense mutation did not seem to hamper the protein amount and the assembly of CV.”
How is the expression in the other tissues? You didn ́t include other tissues in your study. Can you argue about this?
We apologize to the reviewer for the lack of clarity here. In order not to incur in overstatements, since the expression level was not compared to other tissues, and the aim here was to verify whether the mutation may affect protein synthesis and CV assembly, we rephrased as follows: Based on a positive MT-ATP6 immunohistochemical staining (Fig. 3G), the missense mutation did not seem to cause an impairment in the subunit synthesis, and complex V (CV) resulted fully assembled (Fig. 3H).
Response: A non-cancer ovarian tissue should be added to compare.
Response: We agree with the reviewer that a non-cancer tissue is proper for IHC. However, the normal tissue is scarce and seldom available in terms of a proper amount of starting material. Hence, we finally decided to leave out the IHC datum on ATP6 as this has been surpassed by the results obtained by Blue Native Page, which is the most appropriate technique to determine the respiratory complexes assembly. In Figure 3H (now 3G) it is evident that CV is equally assembled in the three samples despite the presence or not of the mutation in MT-ATP6 (with no need for a non-cancer control whose material is not sufficient to extract mitochondria). This suggests that the effect of such mutation does not impair the steady state levels of fully assembled CV although we can not exclude its possible functional impact on the enzyme activity. The Results text has been modified accordingly.
16.- In general the blots should be checked again “The photographs do not correspond to the same blots, therefore I recommend to present the correct ones. The bands from vinculin are oppose to LDHA in the last lane therefore I can assume that photographs are not from the same blot.”
We apologize to the reviewer for overlooking this, as we reported the vinculin taken from a second blot with the same loading. The vinculin from the same blot of LDHA has now been reported in Fig. 3C.
Response: Blot A as it can be seen the bands runs similar and vinculin and S6K can said that are from the same membrane. However, in Blot B Vinculin looks right while in PKM2 an inclination can be appreciate since SCCOPT continuing and being more pronounce in HGSC line. In the same concern it can be seen that vinculin of the upper panel has a up inclination in the right corner while the band of PKM1 do not has it. I recommend to check in deep your blots to assure that are correct. Actually, I recommend to send the complete Blot with out cutout.
Response: The reviewer will understand that unless gels are pre-cast (and not even in all such cases), bands may run not as straight as one would wish for a perfect blot. This occurs especially at the edge of gels due to currents, hence, in the same gel, bands at lower MW slightly bend in a ‘bell’ shape as the front of the loaded protein proceeds down. This is so common especially with home-made gels, that we found this remark quite surprising, having assured the reviewer that the blots have been correctly exposed in the paper and re-checked for revision round 1. The vinculin bands in Fig 3B are indeed the loading controls for PKM1 and PKM2 respectively, blotted on the same membranes previously cut for a more efficient antibody binding and to avoid non-specific cross-reaction.
17.- In the electropherogram I observed a difference in pick intensity, could this be an error of the reaction? Additionally, the image of Perit Met and OPSmCC are identical. Why HGSC was not included?
We are not sure we understand here what the reviewer is referring to. Whether to the picture intensity in terms of color, or the peaks height (which does not reflect intensity). If the reviewer is referring to the different height of the peaks, this is random within every electropherogram in Sanger sequencing (and due, for instance, to presence of residues of ethanol after washing plates) but does not affect the base call. Errors of reaction, moreover, are ruled out on a different DNA extraction, PCR and sequence, which in this case were performed on the peritoneal metastasis, as well as on the primary tumor, revealing the same mutation, which did not occur in the non-cancer matched tissue. This indicated the mutation to be cancer-specific, confirming the peritoneal metastasis originated by a mtDNA mutated clone of the primary, with no need for a non-matched tumor control (HGSC).
Response: If you don’t analyze HGSC you cannot conclude that is a cancer-specific mutation, because it can be a general cancer-mutation, therefore, I recommend to add the HGSC sequencing sample.
Response: While we agree that nuclear mutations may be general cancer-mutations, see for instance all oncogenic hotspots, this is not applicable to mtDNA mutations, as the bulk of literature in the past decade has shown, to which these authors have specifically and substantially contributed. Our definition of cancer-specific mutation refers to comparison with the normal tissue of the same patient and is widely used instead of ‘somatic’, which has the same meaning: we do not envision what the issue could be in this case. No general cancer-mtDNA mutations are nowadays recognized in analogy to, for instance, BRAF or p53 mutations, as they are mostly passenger events that may behave either as bystanders or as phenotypic modifiers if they accumulate to homoplasmy and are not purified by selection (which is almost always the case, particularly if they are disruptive, such as in oncocytic tumors).
On the other hand, a few somatic mtDNA mutations may occur in more than one cancer sample worldwide, which is what we assume the reviewer is asking here: whether this is the case for our mutation. We realize this may not be deduced in a straight-forward fashion from our Methods and Results, where we describe the mutation as previously reported in a genetic disease only (as a germ-line event in that case). Sequencing a single cancer case, when thousands of mtDNAs from cancers are available in the databases to compare would lead, in our opinion, to a biased conclusion, which we avoid in most of our studies by exploiting all mtDNA databases (both of normal and cancer tissues – this is also thanks to our affiliation to the global MSeqDR consortium for annotation of mtDNA variations (Falk MJ et al. Mitochondrial Disease Sequence Data Resource (MSeqDR): a global grass-roots consortium to facilitate deposition, curation, annotation, and integrated analysis of genomic data for the mitochondrial disease clinical and research communities. Mol Genet Metab. 2015)). This should have been extrapolated by our reference, particularly from the Santorsola et al. and from the Preste et al references, but we apologize if this was not the case.
To improve clarity, we now detailed the Methods, by maintaining the former references but detailing all databases that were consulted to deem the mutation as “never reported in somatic cancer tissues before”, a sentence that has been introduced in the Results to complement the previous description in the Charcot case. In order not to miss the most recent reports, we re-consulted the latest updates of the databases for this second round of revision, and additionally checked the most authoritative datasets on cancer specifically, published in Ju et al. Origins and functional consequences of somatic mitochondrial DNA mutations in human cancer, eLife 3:e02935, 2014, and Yuan, Y., et al. Comprehensive molecular characterization of mitochondrial genomes in human cancers. Nat Genet 52, 342–352, 2020, whereby the current revision date of consultation is reported in the Methods.